# Extrahepatic Vitamin K-Dependent Gla-Proteins–Potential Cardiometabolic Biomarkers

**DOI:** 10.3390/ijms25063517

**Published:** 2024-03-20

**Authors:** Bistra Galunska, Yoto Yotov, Miglena Nikolova, Atanas Angelov

**Affiliations:** 1Department of Biochemistry Molecular Medicine and Nutrigenomics, Medical University of Varna, 9000 Varna, Bulgaria; miglena.todorova@mu-varna.bg; 2First Department of Internal Diseases, Section Cardiology, Medical University of Varna, 9000 Varna, Bulgaria; yoto.yotov@mu-varna.bg (Y.Y.); atanas.a.angelov@mu-varna.bg (A.A.); 3Second Cardiology Clinic, Department of Non-Invasive Cardiology, University Hospital “St. Marina”, 9000 Varna, Bulgaria; 4First Cardiology Clinic with Intensive Cardiology Activity, University Hospital “St. Marina”, 9000 Varna, Bulgaria

**Keywords:** vascular calcification, cardiometabolic, Gla-proteins, osteocalcin, matrix Gla-protein, Gla-rich protein, calcification inhibitors

## Abstract

One mechanism to regulate pathological vascular calcification (VC) is its active inhibition. Loss or inactivation of endogenic inhibitors is a major inductor of VC. Such inhibitors are proteins rich in gamma-glutamyl residues (Gla-proteins), whose function strongly depends on vitamin K. The current narrative review is focused on discussing the role of extrahepatic vitamin K-dependent Gla-proteins (osteocalcin, OC; matrix Gla-protein, MGP; Gla-rich protein, GRP) in cardio-vascular pathology. Gla-proteins possess several functionally active forms whose role in the pathogenesis of VC is still unclear. It is assumed that low circulating non-phosphorylated MGP is an indicator of active calcification and could be a novel biomarker of prevalent VC. High circulating completely inactive MGP is proposed as a novel risk factor for cardio-vascular events, disease progression, mortality, and vitamin K deficiency. The ratio between uncarboxylated (ucOC) and carboxylated (cOC) OC is considered as an indicator of vitamin K status indirectly reflecting arterial calcium. Despite the evidence that OC is an important energy metabolic regulator, its role on global cardio-vascular risk remains unclear. GRP acts as a molecular mediator between inflammation and calcification and may emerge as a novel biomarker playing a key role in these processes. Gla-proteins benefit clinical practice as inhibitors of VC, modifiable by dietary factors.

## 1. Introduction

Cardiovascular diseases (CVD) are the leading cause of death and disability in Europe and worldwide. The leading diseases are coronary heart disease (CHD) and stroke. Deaths from CVD exceed those from all cancers in absolute numbers [1]. These diseases lead to excess premature mortality before the age of 70, with CVD accounting for over 60 million potentially lost young lives in Europe [2]. Ischemic heart disease (IHD) was the leading cause of death in 2023 in the USA [3], as well as in other countries, such as Canada and New Zealand [4]. Every one in three deaths was due to these diseases in the United States in 2001–2011 [5], despite the 31% recorded reduction in mortality from atherosclerotic CVD. The situation in Europe is similar, with some regional variability. Since 1980, there has been a favorable downward trend in CVD mortality in Northern, Western and Southern Europe. In Central and Eastern Europe, however, the curve was horizontal or upward in different countries during the same time period [6]. The age-adjusted CVD mortality rate in the region is about twice the EU or economically advanced European average. For example, in Latvia and Romania, CVD mortality rates are 883 and 951/100,000 inhabitants, respectively, and are more than twice the EU average of 373.6/100,000 inhabitants. This is also true for premature mortality. In Russia and Belarus it is more than 10 times higher than in Switzerland—300 vs. 26 per 100,000 inhabitants. CVD mortality rates for men aged 55–59 in some of the countries of the former Soviet Union are higher than those for men aged 75–79 in France [5]. The negative demographic trends with ageing of the population predetermine an increase in CVD morbidity and the associated health care costs [6]. For this reason, prevention of atherosclerotic CVD continues to be a major imperative for societies. The impact analysis of coronary heart disease (CHD) mortality trends in the United States from 1980 to 2000 indicates that at least 44% of the reduction in CVD mortality was due to risk factor (RF) modification in the general population [7]. The same results were reported in other countries [8].

Coronary heart disease has a long asymptomatic period, which provides an excellent opportunity for early preventive interventions on multiple risk factors. Identification of individuals at increased CVD risk is a prerequisite for successful risk reduction. Recently, in addition to traditional preventive measures, such as lowering blood pressure and cholesterol levels, healthy lifestyle-diet, smoking cessation, and sufficient physical activity, particular attention has been paid to the establishment of molecular biomarkers for early detection and monitoring of the course of CVD and its complications. 

Over 90% of atherosclerotic plaques undergo calcification. Aortic calcification is found in 65% of people in the general population over the age of 60 years and correlates with coronary calcification with a positive predictive value for increased CV morbidity and mortality in asymptomatic intermediate-risk patients [9,10]. 

Recently, particular attention has been given to establish molecular biomarkers for early detection and monitoring of the course of CVD. One mechanism to regulate pathological vascular calcification has been shown to be the active inhibition by certain molecules. The loss or inactivation of these endogenic inhibitors is one of the major mechanisms in the genesis of ectopic calcium deposition in vascular walls. Such inhibitory molecules are proteins rich in gamma-glutamyl residues (Gla-proteins), whose mechanism of action, the role in regulatory processes affecting calcium deposition in the vascular wall, has been intensively studied in recent years [11].

Among the multiple inhibitors of vascular calcification, the subject of this review, are the extrahepatic Gla proteins, whose functional activity strongly depends on the vitamin K level. This is important because vitamin K status appears to be a modifying factor with respect to the inhibitory activity of these Gla proteins. Thereof, a subject of the present review, are only those extrahepatic vitamin K-dependent Gla proteins that are related to ectopic calcification such as vascular calcification. More in-depth studies and data in the literature are to be found for osteocalcin, matrix Gla-protein, Gla-rich protein.

Therefore, the current narrative review is focused on discussing the role of extrahepatic vitamin K-dependent Gla-proteins (osteocalcin, matrix Gla-protein, Gla-rich protein) and their relationship with CVD pathology, the degree of arterial calcification, and the interplay between their circulating forms and the conventional metabolic and etiologic risk factors for CVD.

## 2. Pathophysiology and Pathogenesis of Vascular Calcification

One of the early manifestations in the pathogenesis of atherosclerosis and an independent risk factor for cardiovascular complications such as stroke, myocardial infarction, coronary heart disease and mortality is vessel calcification [12]. Over 90% of atherosclerotic plaques undergo calcification. For example, aortic calcification is found in 65% of individuals in the general population aged 60 years and older and correlates with coronary calcification with a positive predictive value for increased cardiovascular morbidity and mortality in asymptomatic intermediate-risk patients [9].

Vascular calcification (VC) may occur in the intima or in the medial part of the arterial wall [13]. In the intima, the fibrous atherosclerotic plaque may include calcium early in the development of the disease, as it has been proven by intravascular ultrasound imaging [14]. The early phase is microcalcification which is difficult to detect. The medial calcification usually occurs in the large arteries and results in the atherosclerosis of the smaller elastic arteries. This type of calcification is more often found in patients with diabetes, kidney diseases, and elderly people. Other risk factors may be arterial hypertension, especially with non-dipping status [15], dyslipidemia [16], abdominal fat composition. Certain genetic conditions can enhance the ectopic calcium deposition in the vessels, such as pseudoxanthoma elasticum, generalized arterial calcification of infancy, and others [17].

The concept that vascular calcification is a passive process has now been revised and it is considered to be an active, cell-mediated, and regulated multifactorial phenomenon that arises as a result of an imbalance between mechanisms that stimulate calcium deposition in the vascular wall and those that inhibit it [18,19]. However, the underlying molecular mechanisms of VC have not been fully elucidated. It is considered that the major mechanism in the development of VC is the osteogenic differentiation of VSMCs into osteoblast-like cells [20,21]. In the course of this process, VSMCs transdifferentiate into osteoblast-like cells and express the master transcription factor of osteogenesis Runx2, as well as other osteogenic factors such as alfa-SMA (α-smooth muscle actin), SM-MHC (smooth muscle myosin heavy chain), alkaline phosphatase (ALP), Runx2 (runt-related transcription factor 2), SOX9 (Sry-related HMG box 9), Msh Homeobox 2 (MSX2), Osterix, osteocalcin and bone sialoprotein II [22]. Thus, differentiated VSMC cells generate matrix vesicles lacking calcification inhibitors, apoptotic bodies that appear to be centers for calcium phosphate precipitation, and as a consequence a mineralization process with the formation of bone-like matrix in the arterial wall is initiated [20,23,24]. Moreover, transdifferentiated VSMCs lose their ability to produce mineralization inhibitors, and secrete osteogenic proteins such as osteocalcin, type I collagen, BMP-2, and alkaline phosphatase [25]. Although the major factor that activates VSMC osteoblastic differentiation is the transcriptional regulator Runx2, many other factors also influence their phenotypic transformation. These include inducers of vascular calcification such as chronic stressors, ionic and metabolic disorders, inflammation, hormonal imbalance, oxidative stress, high phosphate levels, bone morphogenic protein-2 (BMP-2), fibroblast growth factor 23 (FGF-23)-Klotho, alkaline phosphatase, vitamin K deficiency or vitamin K antagonists, as well as endogenous inhibitors like fetuin-A, matrix-Gla protein (MGP), osteopontin (OPN), osteoprotegerin (OPG), BMP-7, and pyrophosphate [18,19,26,27,28]. Osteo/chondrogenic transformed VSMCs secrete inflammatory cytokines like tumor necrosis factor alfa (TNF-alfa), IL-1beta, and IL-17A that in turn can also induce phenotypic VSMC transformation and consequent calcification via upregulation of Wnt/beta-catenin signaling pathway. Thus, a vicious cycle is formed which contributes to the deepening of VC [29,30] and the altered VSMC phenotype and the imbalance between activators and inhibitors of vascular calcification create a microenvironment favoring the initiation and progression of VC.

## 3. Gla-Proteins Are Regulatory Molecules of Ectopic Calcification

Gla-proteins are vitamin K-dependent proteins and are of particular interest as inhibitors of vascular calcification, both because of the widespread use of indirect anticoagulants and their associated non-hemostatic complications, and because of the possibility that circulating levels of their active functional forms could be increased by therapeutic intervention and/or diet.

There are currently 17 known types of vitamin K-dependent proteins, including blood coagulation factors, matrix Gla-protein (MGP), growth arrest specific protein 6 (Gas6), anticoagulant proteins C, S, and Z, osteocalcin (OC), Gla residue-rich protein (GRP), periostin (isoforms 1 and 4), periostin-like factor (PLF), proline-rich Gla proteins (PRGP 1 and 2), transmembrane Gla proteins (TMG3 and TMG4). Only 14 of them have been identified in humans [31,32,33]. The biosynthesis of coagulation factors and anticoagulant proteins C, S, and Z is in the liver, whereas that of OC, MGP, Gas6, and GRP is in extrahepatic tissues. Of all known vitamin K-dependent Gla-proteins, OC, MGP, GRP, Gas6, periostin, periostin-like factor, and nephrocalcin are beyond the blood coagulation cascade. All Gla-proteins undergo post-translational vitamin K-dependent gamma carboxylation of their glutamate residues [31]. This process converts them to their functionally active, carboxylated forms. The enzyme glutamate gamma-carboxylase, with vitamin K as a co-factor, converts the glutamate (Glu) residues in these proteins to gamma-glutamyl (Gla) [34,35] (Figure 1).

## 4. Matrix Gla-Protein

The discovery of vitamin K-dependent matrix Gla-protein (MGP) as a local tissue inhibitor of vascular calcification has diametrically changed the mechanistic understanding of this process and given way to new search for biomarkers in CVD.

The protective role of active MGP in the pathogenesis of the atherosclerotic process is supposed to be due to two main mechanisms: (1) inhibition of calcium deposition by adsorption on extracellular hydroxyapatite [36]; (2) inhibition of smooth muscle apoptosis by binding bone morphogenetic protein 2 (BMP-2) [37]. A direct relationship between the levels of the two functional forms of MGP—the active form (carboxylated MGP, cMGP) and the inactive form (uncarboxylated MGP, ucMGP)—and vascular mineralization has been demonstrated in experimental animal models. This generated hypothesis about the role of these two forms of MGP as potential biomarkers for vascular calcification applies to humans, as well. The importance of MGP for cardiovascular health is also suggested by the fact that no other alternative mechanism for inhibiting vascular mineralization has been found to be as effective to date [38].

### 4.1. Biosynthesis, Secretion and Functional Forms

Matrix Gla protein belongs to a family of proteins containing gamma-carboxyglutamate residues. It is secreted by vascular smooth muscle cells (VSMCs), fibroblasts, chondrocytes, and endothelial cells while it is also present in the arterial wall, heart, kidney, and lung tissue [39].

Price et al. [40] purified it from bovine bone matrix and initially described this protein in 1983. The authors discovered that it is a protein made up of 84 amino acids, with an approximate molecular weight of 12 kD, which contains five unusual amino acids, designated gamma-carboxyglutamate; therefore the protein was designated as matrix Gla-protein (MGP). MGP contains nine glutamate and five serine residues. Only five of the glutamate residues at positions 2, 37, 41, 47, and 52 undergo vitamin K-dependent gamma-carboxylation [33,41], and three of the serine residues at positions 3, 6, and 9 are phosphorylated. Under physiological conditions, calcitriol stimulates the transcription of MGP gene in VSMCs, whereas retinoic acid represses it. In contrast to other Gla-proteins, mature MGP is not produced by limiting proteolysis as it contains an internal pro-peptide that contributes to its specific properties [42].

MGP has been found to exist in four different conformations as a result of two post-translational modifications: serine phosphorylation and gamma-glutamate carboxylation, the latter of which requires vitamin K as a cofactor. The different functional forms of MGP are distinguished by their activity: p-cMGP (fully active-carboxylated and phosphorylated), dp-ucMGP (fully inactive-dephosphorylated and non-carboxylated), dp-cMGP (partially inactive-dephosphorylated and carboxylated), and ucMGP or p-ucMGP (partially inactive-non-carboxylated and phosphorylated) [38,41]. The concentration of total circulating decarboxylated MGP (t-ucMGP) was 1000-fold higher compared to dp-ucMGP levels. It has been suggested that t-ucMGP consists primarily of the phosphorylated form (p-ucMGP) [39].

### 4.2. Biologic Roles and Mechanism of Action

MGP exerts its function through various physiological mechanisms. Through its negatively charged Gla-residues, it forms a mineralization complex with fetuin-A and phosphate and calcium ions, which inhibits the growth of hydroxyapatite crystals and their accumulation in the vascular walls [43]. Binding to hydroxyapatite crystals, MGP activates macrophages in the arterial wall, resulting in phagocytosis and apoptosis of the MGP-hydroxyapatite complex. The ability of MGP to maintain the normal contractile phenotype of VSMCs is thought to be due to either the direct binding of BMP-2, or by blocking the recognition of BMP-2 by its receptor, thus inhibiting its osteoinductive activity [44]. It seems that MGP acts as a shuttle that transports free calcium from the circulation to the bone [41] (Figure 2).

### 4.3. Total-ucMGP and Vascular Calcification

There is still no clarity regarding the relationship between the different functional forms of MGP, vascular calcification, and CV pathology. 

A recent study investigating the relationship between circulating total-ucMGP (t-ucMGP), the severity of CV pathology, and the degree of arterial calcification assessed by measurement of coronary artery calcium scoring (CACS) in patients with or at high risk for CVD found elevated levels of circulating t-ucMGP in patients with heart failure and atrial fibrillation relative to those without CVD. Depending on the CACS value, circulating t-ucMGP levels increased with increasing arterial calcium, but after excluding patients treated with vitamin K antagonists, circulating t-ucMGP showed an opposite trend; it was lowest in individuals with highest CACS [45].

Several studies found higher t-ucMGP levels in various pathologies associated with cardiovascular complications. Significantly higher plasma levels of t-ucMGP in patients with CACS greater than 100 AU relative to controls without coronary calcium were reported by [46]. Increased levels of circulating ucMGP compared to healthy controls (by 77.1%) were measured in hemodialysis patients [47] and in patients with vascular or osteoarticular diseases [48,49]. Significantly elevated circulating t-ucMGP was found in type 2 diabetic patients with ischemic heart disease [50]. Serum ucMGP levels strongly correlate with the degree of calcification of the abdominal aorta, carotid, and coronary arteries in hypertensive individuals [51].

Other studies found either inverse or no association between serum t-ucMGP, CVD, mortality levels, and the risk for CV events [52,53].

There are also conflicting data regarding changes in circulating t-ucMGP in relation to the type and severity of CVD. Increased circulating t-ucMGP was demonstrated in patients with carotid stenosis [54], arterial stiffness and pulse pressure [55], aortic valve disease [56], ischemic heart disease [50], type 2 diabetes mellitus, chronic kidney disease [47,57,58]. Other studies found no association between plasma MGP and CACS or an inverse relationship between higher plasma t-ucMGP levels and CACS [59,60].

Vitamin K is important for the conversion of the inactive form of MGP to the active form. The Rotterdam study was the first clinical trial demonstrating an association between supplemental vitamin K intake and the reduced risk of coronary heart disease in over 4000 men and women aged 55 years and older without a history of myocardial infarction. It was suggested that vitamin K deficiency might result in an inability to carboxylate vascular MGP, leading to enhanced calcification of atherosclerotic lesions and increased risk of coronary heart disease. Other studies also showed that extremely high levels of circulating t-ucMGP were measured in patients on treatment with vitamin K antagonists [33,61].

### 4.4. dp-ucMGP and Vascular Calcification

As vitamin K-dependent serine phosphorylation is a critical step for the conversion of cMGP to a fully active form, it is suggested that phosphorylated and non-phosphorylated forms of cMGP will have different effects as inhibitors of vascular calcification. The dp-ucMGP accumulates in calcified vessels and its decreased circulating levels are associated with more stable plaques [62]. dp-ucMGP, but not ucMGP, lacks the ability to bind calcium, it is not retained in vascular calcium deposits, and is released into the circulation [63]. This was confirmed by histological studies on carotid plaques of patients who underwent carotid endarterectomy [62]. Significantly higher levels of dp-ucMGP compared to the phosphorylated form were found in myocardial biopsy material from patients with ischemic cardiomyopathy [64]. 

A number of studies on different patient cohorts demonstrated a clear association between circulating dp-ucMGP and vascular calcification. Serum dp-ucMGP correlates with carotid intima–media thickness (cIMT) ratio, carotid–femoral pulse wave velocity (carotid–femoral PWV), with CACS, and with endothelial dysfunction score in healthy postmenopausal women [65], as well as with carotid–femoral pulse wave velocity in patients with type 2 diabetes [66] and with arterial hypertension [67]. According to Mayer et al. [68], in patients with persistent vascular disease, higher serum dp-ucMGP levels are associated with adverse cardiovascular events. High serum dp-ucMGP levels accompanied by both severe vascular calcification and impaired vitamin K status was demonstrated in patients with heart failure [58], chronic kidney disease on hemodialysis [69], renal transplantation [70], and diabetes mellitus [71]. It is believed that dp-ucMGP is a better indicator for vitamin K status than t-ucMGP and plays a key role in the pathogenesis of vascular calcification [72].

### 4.5. MGP and Risk Factors for CVD

A number of studies found correlations between serum MGP and well-known conventional risk factors for CVD such as age, gender, smoking, obesity, hypertension, and hyperlipidemia [41,42,70,72].

A recent study in elderly individuals with CVD of varying severity found significant negative correlations between circulating levels of t-ucMGP and classic CVD risk factors such as arterial hypertension, body mass index (BMI), and waist circumference (WC) as indicators of obesity, lipid metabolism parameters, and serum uric acid, as a novel risk factor [45].

Others reported similar results demonstrating a correlation between t-ucMGP, BMI, and WC [73]. In support of possible involvement of MGP in lipid metabolism is a recent study on animal models demonstrating that MGP is highly expressed in mouse visceral adipose tissue and is possibly involved in the regulation of preadipocyte differentiation [74]. It has been suggested that MGP plays a role in lipid metabolism and serum levels of t-ucMGP are associated with the amount of visceral adipose tissue. 

Elevated serum uric acid (UA) levels are considered as a novel independent metabolic risk factor for CVD [75,76,77]. A limited number of studies examined the relationship between serum UA and circulating MGP. Recently, a significant inverse relation was reported between t-ucMGP and serum UA in CVD patients with arterial calcium deposits [45]. It could be assumed that low plasma t-ucMGP levels together with elevated UA may be a result of worsening cardiovascular disease severity accompanied by extensive vascular calcification leading to ucMGP deposition in the calcified area.

### 4.6. Gene Expression of MGP Protein in Peripheral Blood Mononuclear Cells

Data on gene expression dynamics of MGP protein are scarce and studies have been conducted mainly in biopsy material from various tissues. Peripheral blood mononuclear cells (PBMC) appear to be a suitable model for such studies as they are more readily available biological material obtained by minimally invasive procedures. Since blood cells have contact with all tissues, PBMCs are assumed to reflect their state at any given time. This has been tracked through changes in the levels of iRNAs, microRNAs, and epigenetic modifications in various diseases, including CVD [78,79].

MGP is known to be synthesized and γ-carboxylated in monocytes and T lymphocytes [80]. Infiltration and accumulation of macrophages is associated with early-stage calcification in atherosclerosis, suggesting a link between PBMC and vascular calcification in CVD. The inflammatory process in macrophage infiltration also precedes the osteogenic transformation of VSMC and the release of calcified extracellular vesicles [23]. 

In a recent study on patients with CVD compared to those without CVD but at high risk, the downregulation of MGP gene expression was associated with worsening CVD pathology and the presence of certain risk factors for CVD, such as elevated total- and LDL-cholesterol, Castelli indexes, and UA. MGP expression in PBMC was lower in patients with abdominal obesity, arterial hypertension, and hyperlipidemia [81]. Similar results revealing reduced MGP gene expression were found in interstitial cells of injured aortic valves compared to cells of healthy valves [82], and also in heart failure patients of varying severity [83,84].

Seeking a potential association between MGP expression and several CVD risk factors such as obesity, studies on mouse 3T3-L1 pre-adipocytes found increased levels of both mRNA for MGP and MGP protein itself, following stimulation of adipogenesis [55]. Similar in vitro studies on human pre-adipocytes showed 30-fold increased levels of MGP mRNA after stimulation of adipogenesis [85]. Experimental studies on mice showed that inhibition of MGP activation via suppressed carboxylation as well as with vitamin K antagonists leads to accumulation of adipose tissue [74]. These observations suggest that MGP is also involved in the regulation of adipogenesis and lipolysis while deregulation of its gene expression could be associated with abnormal adipose tissue accumulation and obesity.

The data showing reduced MGP expression, relative to the severity of CVD and related to its risk factors, support the notion that MGP expression in PBMC likely reflects CVD severity. It could be hypothesized that a decrease in newly synthesized MGP with the progression of CVD results in less substrate available for the formation of the active p-cMGP form and hence a higher severity of CVD.

### 4.7. MGP as a Biomarker

The use of a circulating biomarker for CVD and/or vascular calcification is an attractive option. The potential value of such a biomarker in the diagnostic process would be to pre-screen patients prior to performing costly, computed tomographic imaging. There is currently no consensus on the prognostic value of MGP as a clinical biomarker in cardiovascular injury. This may be due in part to differences in the populations included in individual studies in terms of age, type, and severity of CVD, cardiovascular risk factors, and the fact that different analytical methodologies for MGP quantification detect MGP forms with different degrees of phosphorylation and carboxylation [46]. 

MGP is one of the strongest endogenous inhibitors of vascular calcification, the function of which depends on the presence of vitamin K. Its circulating form has no biological activity but reflects the calcification process in the vascular wall and the extent of its inhibition. The circulating levels of MGP depend on the rate of its biosynthesis in the vasculature, its secretion by VSMCs, and on its binding to calcified areas in the vascular wall.

It is unclear which functional forms of MGP circulate in the blood. It was established that dpMGP constitutes a very small fraction of total MGP. Moreover, it has been suggested that the phosphorylated fractions are secreted into the extracellular space, whereas the non-phosphorylated forms are secreted as matrix vesicles and apoptotic bodies. Thus, non-phosphorylated MGP could be a predictor of local VSMC stress. Data from experimental animal models and human studies indicate that circulating ucMGP inversely correlates with CACS. A possible explanation for these facts is the phenotypic change of VSMCs into osteoblast-like cells in response to calcification and the associated reduced biosynthesis of MGP. It could be assumed that low circulating levels of t-ucMGP are an indicator of active calcification due to its accumulation at sites of arterial calcification. As ucMGP reflects its calcium-binding capacity, it is thought that t-ucMGP would have the potential to be a novel biomarker of prevalent vascular calcification [41]. High circulating levels of the completely inactive form dp-ucMGP were implicated as an independent predictor of CVD mortality [86]. Therefore, dp-ucMGP has been proposed as a novel risk factor for CV events, CVD progression, and CVD mortality [86,87]. On the other hand, as dp-ucMGP has no calcium-binding activity and cannot be retained at sites of vascular calcification, serum dp-ucMGP could be used as a marker of vitamin K deficiency and for the extent of vascular calcification [88].

The study of MGP benefits clinical practice as a major inhibitor of vascular calcification, the activity of which can be readily reversed by a diet with increased vitamin K intake [73,89]. Therefore, MGP is being thoroughly investigated as a predictive and prognostic candidate biomarker and as a therapeutic target for the treatment of vascular calcification and CVD.

## 5. Osteocalcin

The exact function of OC in the genesis of vascular calcifications remains incompletely understood. Its carboxylated form (cOC) is vitamin K dependent and biologically active. The non-carboxylated form of OC (ucOC) is a marker of vitamin K deficiency. Levels of OC increase with age; they are elevated in women with osteoporosis and reflect the changes in the bone matrix associated with reduced mechanical strength of bone [90].

### 5.1. Biosynthesis and Secretion

The OC molecule is composed of 49 amino acid residues and has a molecular mass of 5.7 kDa. OC is synthesized and secreted by osteoblasts, odontoblasts, chondrocytes, and osteoblast-like VSMCs. The gene for OC in humans is localized to chromosome 1. In cell signaling ucOC is involved through the G-protein coupled receptors GPRC6A and GPR158. In humans, the expression of the OC gene is activated by the active form of vitamin D3, calcitriol. OC is synthesized as a pro-osteocalcin whose biosynthesis is regulated by activating transcription factor 4 (ATF4). Pro-osteocalcin is converted to mature uncarboxylated OC (ucOC) by limiting proteolysis. The OC molecule contains three glutamate residues at positions 17, 21, 24 of the polypeptide chain, and if they are not carboxylated at the gamma-carbon atom of the glutamate, the OC is in an inactive form, designated uncarboxylated OS (ucOC). Gamma-carboxylation of glutamate residues, involving the vitamin K-dependent enzyme gamma-glutamyl carboxylase, converts the inactive OC into a functionally active protein, carboxylated OC (cOC) [90,91].

The extent of OC carboxylation is regulated by protein tyrosine phosphatase. Incomplete carboxylation of OC forms the third functional form of OC undercarboxylated OC, in which the 17th glutamate residue is most often not carboxylated [92,93,94]. As a component of the bone matrix, OC is stable, but it has a short half-life in the circulation, where it is rapidly degraded by proteases. The predominant form of OC in the circulation is thought to be ucOC, accounting for 40–60% of its total amount. In contrast to cOC, due to the lack of negatively charged carboxyl groups, it is not retained at sites of calcification and thus appears in greater concentrations in the circulation relative to those of cOC [95]. Uncarboxylated OC exhibits hormonal activity by improving glucose tolerance, directly stimulating β-pancreatic cell proliferation and insulin secretion, and indirectly activating the secretion of glucagon-like peptide 1 from the intestine [22,96,97]. It increases insulin sensitivity in liver, muscle, and adipose tissue by stimulating adiponectin secretion, modulating glucose homeostasis by an insulin-independent mechanism [98] (Figure 3).

### 5.2. Osteocalcin, Vascular Calcification and Atherosclerosis

Too little is known yet about the role of OC in vascular calcification and cardiovascular health. Data in the literature are scarce and contradictory [53]. In vitro studies on smooth muscle cells from human aorta incubated in mineralization-inducing medium found no changes in osteogenic marker levels in the presence of physiological and pathological concentrations of ucOC, indicating that ucOC is not actively involved in the process of vascular calcification [95]. Other studies demonstrated that OC levels increase during osteogenic differentiation of mouse chondrocytes and VSMCs. In these cell types, OC induces the expression of osteogenic markers, such as the transcription factors Sox9 and Runx2, as well as type X collagen, alkaline phosphatase, and proteoglycans and activates the mineralization process. In experimental animals it was found that OC can retard nucleation and growth of hydroxyapatite crystals. The fact that OC is detected at sites of calcification at later stages of its initiation suggests that it is not involved in the initiation of the process but in its regulation [99].

Recent studies showed correlations between OC other bone proteins, such as BMP-2, and with atherosclerotic vascular changes evidenced by carotid intima–media thickness and ankle–ankle index [100]. In patients with kidney disease at various stages, circulating OC levels significantly increase with the disease severity but do not correlate with coronary calcium score and calcium density as a measure of arterial rigidity [95]. Epidemiological studies investigating the association between OC and the risk of CV events found lower levels of total serum OC in patients with coronary artery disease [101]. Longitudinal studies report a U-shaped relationship of total serum OC and CVD mortality in men over 70 years of age [102] and in women [103], although with less conclusive data in women. The study of Hu et al. [104] found a correlation between high levels of total serum OC and low arterial pulse in men. It could be concluded that low total OC may be a predictor of CV incident mortality in men [105,106]. In women, higher levels of total serum OC correlated with lower arterial pulse probably due to the increased bone remodeling after menopause, which leads to an increase in circulating total serum OC [104].

Presumably, the initial stage in the development of atherosclerosis is endothelial dysfunction, which is an important prerequisite for the development of CVD. A number of studies investigated the relationship between OC and endothelial and smooth muscle cell function [95,105,107,108,109,110]. While some studies showed an inverse relationship between circulating total OC and CVD [101,111], a large longitudinal study in men aged 70–89 years found a negative correlation between the ucOC/total serum OC ratio and the incidence of myocardial infarction after accounting for conventional risk factors [105,106,112]. A single-center cross-sectional study on men also found a positive correlation between high serum ucOC and coronary artery calcium scoring [113]. Similar data demonstrating a positive correlation between serum ucOC and the incidence of coronary artery disease were found in patients with type 2 diabetes mellitus [114]. A study investigating the relationship between circulating ucOC levels and vascular calcification found that an elevated ucOC level was a positive marker of carotid calcification in patients with arterial hypertension [115]. A study of asymptomatic Korean men found a positive correlation between elevated coronary calcium score and ucOC or ucOC/OC ratio [113]. Recent study on patients with varying CVD severity also found a moderate positive correlation between the ratio of circulating forms of OC (ucOC and cOC) and the levels of serum ucMGP—a marker for vascular calcification in patients without CVD but at high risk for it—in those with proven CVD, and in patients with atrial fibrillation. This significant positive correlation persisted in relation to calcium scoring (CACS 1–99AU and CACS ≥ 100AU) [45]. As the ucOC/cOC ratio is considered as an indicator of vitamin K status, it could be assumed that vitamin K insufficiency or deficiency correlates with arterial calcium levels as assessed by calcium scoring. 

A meta-analysis study concluded that no direct relationship between OC levels, vascular calcification, and/or atherosclerosis was conclusively demonstrated [116]. Other clinical studies examining circulating OC levels in relation to vascular calcification reported a significant inverse relationship between total OC levels, aortic or coronary calcification, carotid intima–media thickness, and calcium scoring [117]. 

These contradictory results suggest that plasma OC levels in different individuals cannot lead to the establishment of a clear causal relationship with CVD risk and CV events. This is partly explained by differences in patients’ baseline characteristics such as racial differences, degree of obesity, and comorbidities, and possibly also in adipokine levels [118,119]. Moreover, it is unclear as to what extent the levels of plasma OC reflect the levels of OC produced by osteoblasts.

### 5.3. Osteocalcin, Glucose Homeostasis, and Cardiometabolic Health

In 2007, Karsenty et al. [120] demonstrated for the first time that ucOC regulates energy metabolism by enhancing insulin secretion from beta-pancreatic cells and improving insulin sensitivity of peripheral tissues. Later, in experimental mouse models, it was shown that ucOC regulates insulin secretion and sensitivity by activating a signaling pathway mediated by the G-protein coupled receptor GPRC6A [96]. The discovery of the regulatory role of OC has raised the question of its involvement in glucose homeostasis and in the metabolic link between bone, skeletal muscle, and fat tissue, and whether OC could be a modulating factor of cardiometabolic health.

In vitro cell culture models and ex vivo studies on muscle tissue established the direct effects of ucOC on myocytes by activating glucose and fatty acid uptake and catabolism [121]. It was suggested that OC does not affect the expression of the glucose transporters GLUT 1 and GLUT 4, but favors the translocation of GLUT 4 to the plasma membrane of myocytes [122,123]. On the contrary, studies on isolated rat adipocytes revealed that both cOC and ucOC enhance insulin sensitivity by increased production of adiponectin and regulate insulin dependent glucose transport [124].

In vivo studies in mice also showed that ucOC affects insulin sensitivity in skeletal muscle [125,126]. It was suggested that the role of ucOC, as a factor improving insulin sensitivity in skeletal muscle in vivo, is rather indirect and dependent on other factors [93,121]. The mechanisms by which ucOC affects insulin secretion in beta-pancreatic cells are thought to involve activation of the ERK–MAPK-signaling cascade, leading to increased beta-cell proliferation and insulin secretion [127,128]. In peripheral tissues, other signaling cascades such as PI3K/Akt, cAMP/CREB, and ER-stress signaling pathways are activated to enhance insulin sensitivity [122,129].

Despite the evidence showing that both OC and ucOC are important energy metabolic regulators, their role on global CV risk in patients with metabolic syndrome remains unclear.

Clinical and epidemiological studies investigating the association between circulating ucOC levels and indicators of metabolic syndrome and diabetes mellitus found that low levels of circulating total osteocalcin (tOC) are associated with an increased risk of type 2 diabetes [130,131], increased body mass index and blood glucose, decreased insulin sensitivity, and increased insulin resistance [132,133,134], as well as a significant negative association with cardiometabolic risk factors such as glycated hemoglobin (HbA1C) levels, high fasting blood glucose, and overweight/obesity risk in patients with type 2 diabetes [118]. Similar results were reported for ucOC, whose circulating levels are inversely correlated with blood glucose and HbA1C levels, insulin resistance, and risk of type 2 diabetes [135,136,137,138,139]. Low plasma ucOC concentrations are inversely correlated with a higher incidence of metabolic syndrome, with serum triglyceride (TG) and HDL-cholesterol levels, with waist circumference, and with arterial hypertension in people above middle age [140]. Given the evidence from these studies, the notion that low levels of circulating OC are a likely predictor of metabolic syndrome and type 2 diabetes has been suggested [141]. Associations between circulating OC and different parameters of metabolic syndrome were tested also by others: increased serum OC correlates inversely with BMI, C-reactive protein (CRP), insulin, and TG and is independently associated with changes in CRP [142]. Lower serum ucOC and cOC are associated with higher waist circumference, triglyceride, glucose, blood pressure, and lower high-density lipoprotein cholesterol [143], with a positive correlation between ucOC and insulin secretion in diabetes patients [144].

A recent longitudinal study on 296 patients with metabolic syndrome showed a significant positive correlation between serum ucOC and HDL-cholesterol and a negative relation with both HbA1c and fasting blood glucose at the 6th and 12th month during follow up. The lower circulating ucOC levels were related to worse metabolic profile and higher cardiovascular risk in patients with methanolic syndrome regardless of the presence of type 2 diabetes. The authors considered that circulating ucOC might be a potential biomarker to both predict type 2 diabetes development and to classify CV risk in patients with metabolic syndrome [145,146].

Based on the assumptions that endothelial progenitor cells expressing OC are able to activate calcification in vitro and in vivo [147], a cross-sectional study in a cohort with coronary artery atherosclerosis evaluated the role of different OC forms. The study revealed a significant association between the count of circulating early endothelial progenitor cells expressing OC, coronary atherosclerosis severity and CV risk factors such as total cholesterol (TC), LDL-C, and TG. Testing the serum levels of ucOC and total OC they found an inverse correlation between ucOC and OC positive early endothelial progenitor cell count, as well as between ucOC and total OC serum levels and the number of CV risk factors [148].

Despite numerous clinical studies confirming the finding that ucOC acts as a hormone and regulates glucose metabolism, the association between ucOC and glucose metabolism remains controversial. A very recent study on mouse models using mice lines with the same genetic deletion of OC, but with a genetic background (phenotype) different from that of Karsenty’s group reported conflicting results regarding the regulatory role of OC on glucose metabolism. Possibly the link between circulating OC and glucose metabolism or CV risk is mediated by physical exercise, which increases bone formation, and as a result OC [90,149].

## 6. Gla-Rich Protein

The newest member of the vitamin K-dependent Gla-protein family is the gamma-glutamate residue-rich protein (GRP). The role and mechanism of action of GRP is still unclear, but it is thought that due to its large number of glutamate residues and its high calcium-binding capacity, it plays a role as a global calcium modulator and as an additional mechanism to that of MGP for inhibiting vascular mineralization [150].

### 6.1. Structure and Functional Forms

GRP was originally identified in sturgeon as a novel vitamin-K-dependent protein. Whereas in sturgeon GRP is found primarily in cartilaginous tissues, in mammals it is found in both skeleton and connective tissue (including bone, cartilage, skin, and vessels) [150]. GRP has a remarkably well conserved Gla domain, which has given rise to the suggestion that it is a novel specific *γ*-carboxylated calcium-binding protein. GRP contains an exceptionally high number of Gla residues, 16 out of 74 amino acids, i.e., 21.6% of the amino acid composition of the protein [151].

GRP exists in two functional forms: carboxylated (cGRP) and uncarboxylated (ucGRP). Similar to MGP and OC, the functionally inactive non-carboxylated form, ucGRP, is increased in diseases associated with pathological calcification such as aortic valve stenosis, osteoarthritis, and neoplasias, while the functionally active carboxylated form is an effective inhibitor of calcification in the cardiovascular system [152,153,154,155].

### 6.2. Biologic Role

In vitro experiments demonstrated that GRP accumulates at sites of pathological calcification and plays a role as a negative regulator of osteogenic differentiation and inhibitor of calcification in blood vessels [154]. It acts also as a modulator of calcium availability in the extracellular matrix as well as a potential inhibitor of calcification in soft tissues. In addition, it is thought that GRP plays a role not only as an inhibitor of pathological vascular calcification, but also as an anti-inflammatory factor in the cardiovascular system, independently of its gamma-carboxylation status [154,155,156]. Moreover, a new nano encapsulated ucGRP formulation was developed and tested in different human in vitro cell systems as an anti-inflammatory agent showing decreased pro-inflammatory response of these cells [157].

Gamma-carboxylation of GRP is essential for its role as a calcification inhibitor [155,158]. Factors such as inadequate dietary vitamin K intake, mutations in the enzyme *γ*-glutamyl carboxylase, and treatment with vitamin K antagonists can lead to reduced *γ*-carboxylation and increased risk of bone fractures or ectopic calcification.

### 6.3. GRP and Vascular Calcification

Data concerning the role of GRP in the mechanism of vascular calcification are very scarce. The carboxylated form of GRP has been found to play a key role in inhibiting ectopic calcification. Given the functional relationship between GRP and BMP-2, it could be hypothesized that GRP is involved in the trans-differentiation of VSMCs into osteoblast-like cells and thus affects vascular calcification [150,152,154]. An in vitro study demonstrated that the calcification process induces GRP expression and accumulation. The authors suggest that GRP is likely a component of the fetuin-A–MGP complex inhibiting calcification in matrix vesicles and calcium-induced signaling pathways, directly binding hydroxyapatite crystals and blocking their accumulation [154].

It is well known that vascular calcification and inflammation are interrelated processes contributing to the development of atherosclerosis. Inflammation is one of the promoters of atherosclerotic changes in the vascular wall and subsequent vascular calcification. There is evidence that reduced GRP levels are associated with increased pro-inflammatory responses, thus confirming the dual role of GRP as a calcification inhibitor and as an anti-inflammatory factor [150,159]. Total serum GRP was found to negatively correlate with serum C-reactive protein (CRP), with CVD severity, and with calcium scoring [160].

Research exploring the association between circulating levels of GRP and CVD severity is extremely limited. A recent study demonstrated a clear trend of decreasing circulating GRP levels in patients with CVD in relation to disease severity [160]. A similar decrease in GRP with increasing severity of cardiovascular pathology was found also by others [154,161,162]. In addition, a study of Viegas et al. [155] found severely reduced serum GRP levels in patients with chronic kidney disease (CKD) with increasing disease severity [155].

It could be assumed, that Gla-rich protein acts as a molecular mediator between inflammation and calcification, playing an important role in the development and progression of complications such as ectopic calcification which are observed in many chronic inflammatory diseases, including CVD and metabolic syndrome [80,154,158]. GRP may emerge as a novel biomarker that plays a key role in the interrelated processes of chronic inflammation and ectopic calcification [158].

## 7. Gla-Proteins and Statins

Statins (inhibitors of 3-hydroxy 3-methylglutaryl coenzyme A reductase (HMG-CoA reductase) play a central role in the treatment of atherosclerosis and coronary heart disease. They are most effective in terms of lowering LDL-cholesterol and slowing the development of atherosclerosis while able to reduce the incidence of major cardiovascular events and stroke by 20% [163,164]. Due to their lipid-lowering and pleiotropic effects, they are a proven agent reducing CV events and the resulting mortality. Statins increase the level of HDL-cholesterol, which has a protective effect against the formation of atherosclerotic plaques, stabilize vascular atheromatous plaques, reduce the possibility of their rupture, and lead to plaque regression. On the other hand, there is evidence that intensive therapy with statins increases vascular calcification and accelerates its progression. To explain these conflicting data on statins, on the one hand as protagonists of calcification and on the other hand as a factor leading to plaque regression and good clinical response, it has been suggested that statins, in addition to having a delipidating effect on plaque, contribute to its transformation and stabilization [165,166,167]. Given the divergent effects of statins on vascular calcification, the question of the relationship between their use, vascular calcification, and the levels of the vitamin K-dependent Gla proteins MGP, OC, and GRP, as potential regulators of this process, is of interest. A recent study on patients with proven CVD or at high risk of CVD found a significant increase in both ucOC levels and the ucOC/cOC ratio in patients on statin therapy [168]. In addition, circulating ucOC levels significantly correlated with coronary calcium score in the entire cohort and in statin-treated patients. The same correlation was not found in those not taking statins. These relationships persisted when multivariate regression analysis was performed, showing that ucOC levels in statin takers are a predictor of high artery calcium score. Elevated levels of ucOC and of ucOC/cOC are an indicator of vitamin K deficiency [53], as demonstrated in patients on statin treatment. It could be speculated that statin-induced vascular calcification is mediated by inhibition of vitamin K-dependent processes, including vitamin K-dependent carboxylation of extrahepatic Gla-proteins. Similar relationships were not found for MGP and GRP [169]. A cohort study in patients with chronic kidney disease found that statins were on the one hand an independent predictor of high CKD and mortality and on the other hand significantly reduced in vitro menaquinone-4 (MK-4) biosynthesis in vascular smooth muscle cells [170]. This may lead to local vitamin K deficiency, suppression of vitamin K-dependent carboxylation, and activation of Gla-proteins mediating the inhibition of vascular calcification [170]. It has been suggested that as inhibitors of HMG-CoA reductase, statins suppress not only the de novo biosynthesis of cholesterol but also of prenyl intermediary metabolites, which in turn leads to reduced conversion of vitamin K1 to menaquinone (MK-4) [171,172]. MK-4, which is the active form of vitamin K2, is produced in vivo by prenylation from vitamin K1 in extrahepatic tissues, including vascular walls. Vitamin K2, as a cofactor of gamma-glutamyl carboxylase, is required for the activation of extrahepatic Gla-proteins. Thus, by affecting the in vivo formation of MK-4, statins could impair the activation of Gla-proteins such as MGP, OC, and GRP, which are potential inhibitors of vascular calcification.

## 8. Future Directions

The degree of calcium deposition in coronary vessels is perceived as a marker of the biological age of the vessels and the determination of CACS is a modern and up-to-date method for objective assessment of the risk of cardiovascular events. In addition to the determination of CACS, which involves an invasive examination of the vascular status, a wide range of Gla-proteins, such as MGP, OC, and GRP are an attractive field of study as endogenic inhibitors of vascular calcification. Demonstration of a causal relationship between the levels of different functional forms of these Gla-proteins and CACS, as well as its validation in large cohorts of patients with CVD would be the rationale for their future inclusion as non-invasive molecular biomarkers to assess the presence of atheromatous vascular deposits and especially their degree of calcification or to determine the risk of vascular calcification. In this aspect, different functional forms of MGP are being thoroughly investigated as predictive and prognostic candidate biomarkers and as a therapeutic target for the treatment of vascular calcification and CVD.

A particular characteristic of these proteins is that their functionally active forms are dependent on vitamin K. Proving the diagnostic utility of different functional forms of these Gla-proteins will enable future studies that could change the diagnostic algorithm by introducing circulating vitamin K-dependent Gla-proteins to other biomarkers already established and used in practice, as well as to change the treatment algorithm by including vitamin K supplementation of CVD patients.

Moreover, modulation of extrahepatic Gla-protein activity by vitamin K is essential for clinical practice when anticoagulant treatment is necessary that should be conducted with direct anticoagulants (DOAC) that do not affect the function of vitamin K-dependent gamma-carboxylases and the activation of these endogenic VC inhibitors. As it is not yet studied how DOAC will affect different functional forms of Gla proteins and the process of vessel calcification, this is another important aspect of future studies. As the uc-dpMGP and ucOC/cOC ratio has been found to reflect the body’s extrahepatic functional vitamin K status, validation of these parameters in large cohort studies would allow the evaluation of the effects of vitamin K supplementation on cardiovascular outcomes. This will subsequently contribute to the development of an adequate program to monitor functional vitamin K status and assess the need for supplementation with vitamin K2 or long-chain menaquinones to reduce the complications of prevalent CVD.

Another point of future studies that is still obscure and needs further clarification, is related to possible molecular mechanisms through which statins, a widely used medication in the treatment of atherosclerosis and coronary heart disease, may affect vitamin K status, the activity of vitamin K dependent Gla-proteins, and their functions involved in vascular protection.

Regarding the link with the CVD risk profile, the evidence for the univariate effect of Gla-proteins in CVD development are persuasive, but future studies are needed to prove that they retain their independent effect in multivariate analyses and to show their additive effect on the CVD risk profile.

As vascular calcification is involved in the pathogenesis of numerous diseases, it could be assumed that active forms of vitamin K-dependent Gla proteins also play a role in other diseases beyond CVD. Matrix Gla proteins may be useful biomarkers in other pathological conditions. Although there are some data on their presence in patients with type 2 diabetes mellitus [173,174], the research is sparse on people with type 1 diabetes. Those patients are at excessive risk of micro- and macrovascular complications and of early atherosclerosis and calcification of the arteries.

The COVID-19 pandemic gave a boost to the research on the pathogenesis of vascular reactions and alterations in infectious diseases. The vitamin K metabolism and depletion during COVID-19, together with the decrease in the vitamin K-dependent activation of MGP leaves the elastic fibers in the hepar unprotected against infection, with increased thrombogenicity [175]. Future efforts should be concentrated on the pathophysiology of the MGP in other acute infections, as well as in more chronic conditions, like tuberculosis. It may reveal their role in the development and adaptation of the immune system.

Initial studies demonstrate high dp-ucMGP levels in COPD patients. Vitamin K deficiency leads to loss of elastin in lung tissues [176]. There is a correlation between lung emphysema and vitamin K status, with increased levels of dp-ucMGP and diffuse capacity for carbon monoxide. Epidemiologic studies revealed higher mortality in patients with COPD in the upper quartile of dp-ucMGP [177]. However, the exact mechanisms and the diagnostic work-up are still not very clear and need further research.

Another possible application of the MGPs is in chronic inflammatory diseases. There have been preliminary studies in rheumatoid arthritis [178], chronic inflammatory colitis, and other diseases. The future perspective is for more extensive research in autoimmune diseases and other chronic inflammation conditions which may reveal the potential role of vitamin K related proteins in the pathologic processes.

## 9. Conclusions

Vascular calcification is inextricably linked to atherosclerotic vascular disease, follows the course of the atherosclerotic process, and also increases with age. In addition to evaluating CACS, non-invasive methods including biomarkers are being sought to assess the presence of atheromatous vascular deposits and especially their degree of calcification or to determine the risk of vascular calcification. A wide range of Gla-proteins, such as MGP, OC, and GRP belong to an attractive field of study as endogenic inhibitors of vascular calcification. A particular characteristic of these proteins is that their functionally active form is dependent on vitamin K. This is essential for clinical practice, not only because of the widespread use of indirect anticoagulants as vitamin K antagonists, but also because the circulating levels of the active functional forms of Gla-proteins can be increased by dietary modification (food intake, vitamin K supplementation). If anticoagulant treatment is necessary, then it should be conducted with direct anticoagulants that do not affect the function of vitamin K-dependent gamma-carboxylases. The Gla-proteins and especially different functional forms of MGP are being thoroughly investigated as predictive and prognostic candidate biomarkers and as a therapeutic target for the treatment of vascular calcification and CVD and with a potential for other pathologic conditions.

## Figures and Tables

**Figure 1 ijms-25-03517-f001:**
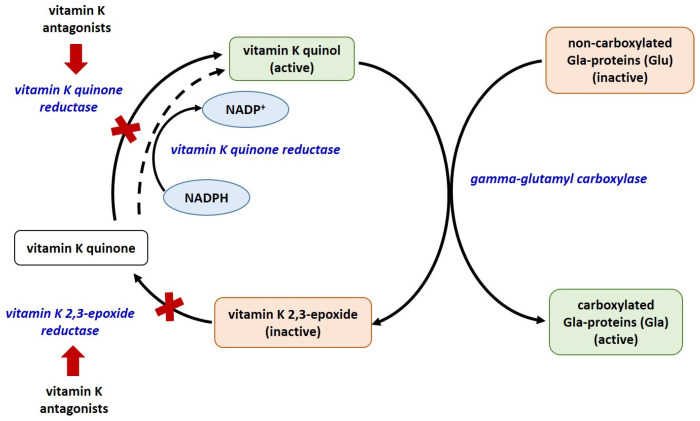
The vitamin K cycle. Red arrows – inhibition; red X—blocked process; all enzymes are given in blue colored font.

**Figure 2 ijms-25-03517-f002:**
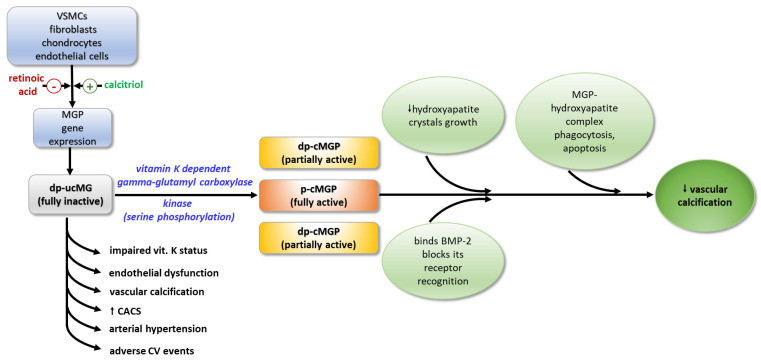
Activation and biological effects of MGP. MGP—matrix Gla-protein; dp-ucMGP—fully inactive dephosphorylated and non-carboxylated MGP; dp-cMGP—partially inactive dephosphorylated and carboxylated MGP; p-cMGP—fully active carboxylated and phosphorylated MGP; BMP-2—bone morphogenic protein-2; VSMCs—vascular smooth muscle cells; CACS—coronary arterial calcium score.

**Figure 3 ijms-25-03517-f003:**
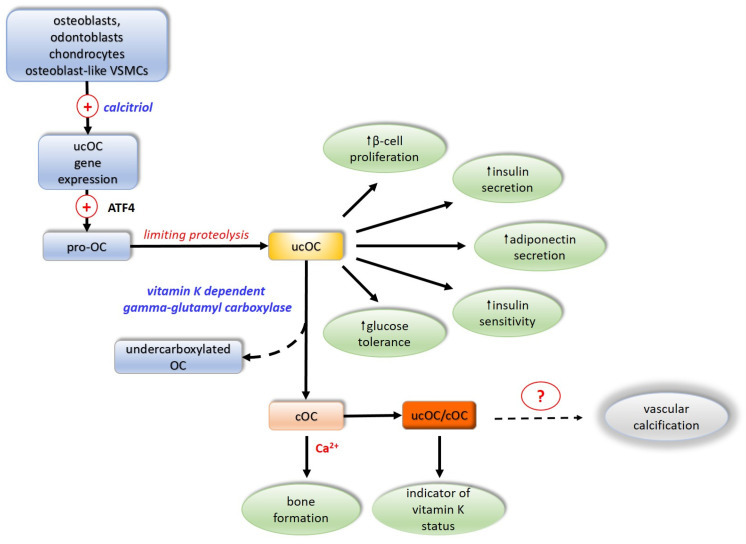
Activation and biologic effects of osteocalcin. ATF4—activating transcription factor 4; OC—osteocalcin; ucOC—inactive non-carboxylated osteocalcin; cOC—active carboxylated osteocalcin; VSMCs—vascular smooth muscle cells. **(+)**—activation; dotted arrows—unclear mechanism; **Ca^2+^**—calcium ions; ?—not firmly proved involvement.

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
