# Peer review of "Extrahepatic Vitamin K-Dependent Gla-Proteins–Potential Cardiometabolic Biomarkers"

_ijms, 2024, doi:10.3390/ijms25063517_

Round 1

Reviewer 1 Report

Comments and Suggestions for Authors

-       This is clearly a narrative review. Therefore, this point can be clearly stated and there is no need to discuss limitations, since the limitations of narrative reviews are well-know.

-       Similarly, it makes no sense to include a section for the methods, since this is not a systematic review.

-       A specific section for “perspectives/future direction” is appropriate, but should not be merged with the conclusion. In this section, the authors could also highlight and discuss the current gaps in the knowledge of this specific topic, which could be the “limitations” of the available evidence (rather than limitation of a narrative review)

-       Conversely a conclusion section consisting of a paragraph is important to highlight the main relevance of these molecules according to the available literature.

-       I would also suggest to add a section discussion discussing the role of Gla-proteins in other diseases.

-       In this additional section, a mention to COVID-19 should be also done, considering the predisposing role of CVD for severe forms of this disease.  Therefore, it would be interesting to check and discuss if there is any evidence in this regard, as well. This article (e.g. The Pleiotropic Role of Vitamin K in Multimorbidity of Chronic Obstructive Pulmonary Disease. J Clin Med. 2023 Feb 5;12(4):1261. doi: 10.3390/jcm12041261) and others may provide useful insights in this regard. Moreover, a recent paper reported that MGP can facilitate CD8+ T cell exhaustion by activating the NF-κB pathway (see: Int J Biol Sci. 2022 Mar 6;18(6):2345-2361. doi: 10.7150/ijbs.70137) and CD8+ T cell subpopulation balance/homeostasis has been shown to be significantly perturbated in COVID-19 patients (refer to: Respir Res. 2022 Oct 10;23(1):278. doi: 10.1186/s12931-022-02190-8).

-       “There are currently 17 known types of vitamin K-dependent proteins, including  blood coagulation factors, matrix Gla-protein (MGP), growth arrest specific protein 6  (Gas6), anticoagulant proteins C, S and Z, osteocalcin (OC), Gla residue-rich protein  (GRP), periostin (isoforms 1 and 4), periostin-like factor (PLF), proline-rich Gla proteins (PRGP 1 and 2), transmembrane Gla proteins (TMG3 and TMG4). Only 14 of them have been identified in humans [8, 9, 10].” The authors mentioned that there are at least 14 vit. K dependent-proteins expressed in humans. However, specific sections were dedicated to few of them. The authors should clarify why only these were described in detail and, in general, at the end of the introduction should better explain the objective of this narrative review.

Comments on the Quality of English Language

See above

Author Response

BISTRA GALUNSKA

Department of Biochemistry, Molecular medicine and Nutrigenomics

Medical University of Varna

9000 Varna, Bulgaria

[email protected]

To

PROF. DR. DARIUSZ SZUKIEWICZ

Guest Editor

Special Issue “Molecular Mechanisms of Action of Adaptogens -

in Search of Natural Methods of Restoring Homeostasis”

MDPI

CC: Reviewer 1 of Manuscript ID: ijms-2901141

Type of manuscript: Review

Title: Extrahepatic vitamin K-dependent Gla-proteins – potential

cardiometabolic biomarkers

Authors: Bistra Galunska*, Yoto T Yotov, Miglena N Nikolova, Atanas A Atanasov

March 08, 2024

Dear Prof. Dr. Szukiewicz,

Dear Reviewer,

On 1-th March 2024, we were announced by email that our manuscript (ijms-2901141) entitled " Extrahepatic vitamin K-dependent Gla-proteins – potential cardiometabolic biomarkers", submitted for consideration for publication in the International Journal of Molecular Sciences, Special Issue “Molecular Mechanisms of Action of Adaptogens – in Search of Natural Methods of Restoring Homeostasis” as an review article has been reviewed.

We would like to thank the Reviewer for the critical remarks and recommendations. All required revisions are included in the abstract, main text, and reference list. The changes we made in the manuscript are highlighted in yellow text.

Below are the detailed answers point by point to the reviewers’ comments. All answers and explanations are highlighted using bold and blue coloured text.

Comments and Suggestions for Authors

Reviewer 1

  1. This is clearly a narrative review. Therefore, this point can be clearly stated and there is no need to discuss limitations, since the limitations of narrative reviews are well-know.

Answer: In the revised manuscript, we have clearly stated that this is a narrative review.

We consider that it is appropriate to point out some limitations of the review, as the literature data concerning some aspects of the role of Gla-proteins in vascular calcification are very limited and obviously cannot be discussed.

  1. Similarly, it makes no sense to include a section for the methods, since this is not a systematic review.

Answer: Thank you for this comment. We believe that the inclusion of a section, indicating the main descriptors in literature search, (criteria for the selection of literature sources, keywords used, etc.) is useful.

  1. A specific section for “perspectives/future direction” is appropriate, but should not be merged with the conclusion. In this section, the authors could also highlight and discuss the current gaps in the knowledge of this specific topic, which could be the “limitations” of the available evidence (rather than limitation of a narrative review)

Answer: The section “Future directions” is separated from section “Conclusions”. Current gaps in the literature regarding the role of discussed Gla-proteins in cardiometabolic health are pointed.

  1. Future Directions

There is a bulk of evidence of the participation of vitamin K antagonists (VKA) in the vascular calcification. During the last decade, the use of direct oral anticoagulants (DOAC) is rising exponentially and is replacing the VKA for many indications. It is not clear how they will affect the Gla proteins and the process of vessel calcification.

The evidence for the univariate effect of the Gla-proteins in the CVD development are persuasive but there are not enough studies to prove that they retain their independent effect in multivariate analyses and to show their additive effect on the CVD risk profile.

  1. Conversely a conclusion section consisting of a paragraph is important to highlight the main relevance of these molecules according to the available literature.

Answer: The conclusion section has been edited accordingly. The inserted/edited text is highlighted in yellow.

  1. Conclusions

Vascular calcification is inextricably linked to atherosclerotic vascular disease, follows the course of the atherosclerotic process and also increases with age. It is no coincidence that the degree of calcium deposition in coronary vessels is perceived as a marker of the biological age of the vessels and the determination of CACS is a modern and up-to-date method for objective assessment of the risk of cardiovascular events.

In addition to the determination of CACS, which involves an invasive examination of the vascular status, additional non-invasive methods are being sought to assess the presence of atheromatous vascular deposits and especially their degree of calcification or to determine the risk of vascular calcification. A wide range of Gla-proteins, such as MGP, OC, and GRP are an attractive field of study as endogenic inhibitors of vascular calcification. A particular characteristic of these proteins is that their functionally active form is dependent on vitamin K. This is essential for the clinical practice, not only because of the widespread use of indirect anticoagulants as vitamin K antagonists, but also because the circulating levels of the active functional forms of Gla-proteins can be increased by dietary modification (food intake, vitamin K supplementation) or if anticoagulant treatment is necessary, it should be conducted with direct anticoagulants that do not affect the function of vitamin K-dependent gamma-carboxylases. Therefore, these Gla-proteins and especially different functional forms of MGP are being thoroughly investigated as predictive and prognostic candidate biomarkers and as a therapeutic target for the treatment of vascular calcification and CVD.

  1. I would also suggest to add a section discussion discussing the role of Gla-proteins in other diseases.

In this additional section, a mention to COVID-19 should be also done, considering the predisposing role of CVD for severe forms of this disease.  Therefore, it would be interesting to check and discuss if there is any evidence in this regard, as well. This article (e.g. The Pleiotropic Role of Vitamin K in Multimorbidity of Chronic Obstructive Pulmonary Disease. J Clin Med. 2023 Feb 5;12(4):1261. doi: 10.3390/jcm12041261) and others may provide useful insights in this regard. Moreover, a recent paper reported that MGP can facilitate CD8+ T cell exhaustion by activating the NF-κB pathway (see: Int J Biol Sci. 2022 Mar 6;18(6):2345-2361. doi: 10.7150/ijbs.70137) and CD8+ T cell subpopulation balance/homeostasis has been shown to be significantly perturbated in COVID-19 patients (refer to: Respir Res. 2022 Oct 10;23(1):278. doi: 10.1186/s12931-022-02190-8).

Answer: Thank you for this recommendation and for extremely interesting papers. At this stage, the focus of our research and of this review is the interrelationship between the functional activity of extrahepatic vitamin K-dependent Gla proteins and vascular calcification as a pathogenic element of cardiovascular events and associated risk factors. For this reason, the role of these proteins in other diseases, including SARS-CoV infection, has not been discussed.

As the extrahepatic vitamin K-dependent Gla proteins are subject of intensive studies in relation to their pleiotropic effects and the possibility of modifying their activity by vitamin K supplementation, we are ready to prepare another review article on the role of extrahepatic Gla proteins in other diseases beyond CVD, which would be of considerable interest to the scientific community.

  1. “There are currently 17 known types of vitamin K-dependent proteins, including blood coagulation factors, matrix Gla-protein (MGP), growth arrest specific protein 6 (Gas6), anticoagulant proteins C, S and Z, osteocalcin (OC), Gla residue-rich protein  (GRP), periostin (isoforms 1 and 4), periostin-like factor (PLF), proline-rich Gla proteins (PRGP 1 and 2), transmembrane Gla proteins (TMG3 and TMG4). Only 14 of them have been identified in humans [8, 9, 10].” The authors mentioned that there are at least 14 vit. K dependent-proteins expressed in humans. However, specific sections were dedicated to few of them. The authors should clarify why only these were described in detail and, in general, at the end of the introduction should better explain the objective of this narrative review.

Answer: We appreciate your recommendation. The following text is inserted at the end of the Introduction: “Among the multiple inhibitors of vascular calcification the subject of this review are the extrahepatic Gla proteins, whose functional activity strongly depends on the vitamin K level. This is important because vitamin K status appears to be a modifying factor with respect to the inhibitory activity of these Gla proteins. Improving vitamin K status by diet or vitamin K supplements, would contribute to reducing the risk of vascular calcification and associated with it adverse outcomes for cardiovascular health. Furthermore, it should be taken into account that when anticoagulant therapy is needed, in order to preserve the inhibitory activity of these vitamin K-dependent Gla-proteins, direct anticoagulants that do not affect the function of vitamin K-dependent gamma-carboxylases are recommended instead of vitamin K antagonists”.

Sincerely,

Bistra Galunska – corresponding author

Reviewer 2 Report

Comments and Suggestions for Authors

Dear authors,

I have carefully studied the manuscript entitled “Extrahepatic vitamin K-dependent Gla-proteins – potential cardiometabolic biomarkers” by Bistra Galunska et al. The topic is very interesting, well structured, with a numerous bibliography. After reading this article, the readers can remain with many information, due to the detailed explanations in all of the Gla-proteins. Thus, there are some minor adjustments that need to be made:

1.     The materials and methods section needs to be the second one, after the introduction.

2.     I would add one more paragraph regarding the pathophysiology of the calcification. I think this would increase even more the quality of the article.

Author Response

BISTRA GALUNSKA

Department of Biochemistry, Molecular medicine and Nutrigenomics

Medical University of Varna

9000 Varna, Bulgaria

[email protected]

To

PROF. DR. DARIUSZ SZUKIEWICZ

Guest Editor

Special Issue “Molecular Mechanisms of Action of Adaptogens -

in Search of Natural Methods of Restoring Homeostasis”

MDPI

CC: Reviewer 2 of Manuscript ID: ijms-2901141

Type of manuscript: Review

Title: Extrahepatic vitamin K-dependent Gla-proteins – potential

cardiometabolic biomarkers

Authors: Bistra Galunska*, Yoto T Yotov, Miglena N Nikolova, Atanas A Atanasov

March 08, 2024

Dear Prof. Dr. Szukiewicz,

Dear Reviewer,

On 1-th March 2024, we were announced by email that our manuscript (ijms-2901141) entitled " Extrahepatic vitamin K-dependent Gla-proteins – potential cardiometabolic biomarkers", submitted for consideration for publication in the International Journal of Molecular Sciences, Special Issue “Molecular Mechanisms of Action of Adaptogens – in Search of Natural Methods of Restoring Homeostasis” as an review article has been reviewed.

We would like to thank the Reviewer for the critical remarks and recommendations. All required revisions are included in the abstract, main text, and reference list. The changes we made in the manuscript are highlighted in yellow text.

Below are the detailed answers point by point to the reviewer comments. All answers and explanations are highlighted using bold and blue coloured text.

Comments and Suggestions for Authors

Reviewer 2

I have carefully studied the manuscript entitled “Extrahepatic vitamin K-dependent Gla-proteins – potential cardiometabolic biomarkers” by Bistra Galunska et al. The topic is very interesting, well structured, with a numerous bibliography. After reading this article, the readers can remain with many information, due to the detailed explanations in all of the Gla-proteins. Thus, there are some minor adjustments that need to be made:

  1. The materials and methods section needs to be the second one, after the introduction.

Answer: An adjustment has been made accordingly. In the revised manuscript the section “Materials and Methods” is the second one.

  1. I would add one more paragraph regarding the pathophysiology of the calcification. I think this would increase even more the quality of the article.

Answer: A new section entitled “Pathophysiology and pathogenesis of the calcification” with corresponding references has been added.

  1. Pathophysiology and pathogenesis of vascular calcification

One of the early manifestations in the pathogenesis of atherosclerosis and an independent risk factor for cardiovascular complications such as stroke, myocardial infarction, coronary heart disease and mortality is vessel calcification [13]. Over 90% of atherosclerotic plaques undergo calcification. For example, aortic calcification is found in 65% of individuals in the general population aged 60 years and older and correlates with coronary calcification with positive predictive value for increased cardiovascular morbidity and mortality in asymptomatic intermediate-risk patients [10].

Vascular calcification (VC) may occur in the intima or in the medial part of the arterial wall [14]. In the intima, the fibrous atherosclerotic plaque may include calcium early in the development of the disease, as it has been proven by intravascular ultrasound imaging [15]. The early phase is the microcalcification which is difficult to detect. The medial calcification usually occurs in the large arteries and result in the atherosclerosis of the smaller elastic arteries. This type of calcification is more often found in patients with diabetes, kidney diseases, and elderly people. Other risk factors may be the arterial hypertension, especially with non-dipping status [16], dyslipidemia [17], abdominal fat composition. Certain genetic conditions can enhance the ectopic calcium deposition in the vessels, such as pseudoxanthoma elasticum, generalized arterial calcification of infancy and others [18].

The concept that vascular calcification is a passive process has now been revised and it is considered to be an active, cell-mediated and regulated multifactorial phenomenon that arises as a result of an imbalance between mechanisms that stimulate calcium deposition in the vascular wall and those that inhibit it [19, 20]. However, the underlying molecular mechanisms of VC are not fully elucidated. It is considered that the major mechanism in the development of VC is the osteogenic differentiation of VSMC into osteoblast-like cells [21, 22]. In the course of this process, VSMC transdifferentiate into osteoblast-like cells and express the master transcription factor of osteogenesis Runx2, as well as other osteogenic factors such as alfa-SMA (α-smooth muscle actin), SM-MHC (Smooth Muscle Myosin Heavy Chain), alkaline phosphatase (ALP), Runx2 (Runt-related transcription factor 2), SOX9 (Sry-related HMG box 9), Msh Homeobox 2 (MSX2), Osterix, osteocalcin and bone sialoprotein II [23]. Thus, differentiated VSMC cells generate matrix vesicles lacking calcification inhibitors, apoptotic bodies that appear to be centers for calcium phosphate precipitation, and as a consequence mineralization process with the formation of bone-like matrix in the arterial wall is initiated [24, 19, 25]. Moreover, transdifferentiated VSMC lose their ability to produce mineralization inhibitors, and secrete osteogenic proteins such as osteocalcin, type I collagen, BMP -2, and alkaline phosphatase [26]. Although the major factor that activates VSMC osteoblastic differentiation is the transcriptional regulator Runx2, many other factors also influence their phenotypic transformation. These include inducers of vascular calcification such as chronic stressors, ionic and metabolic disorders, inflammation, hormonal imbalance, oxidative stress, high phosphate levels, bone morphogenic protein-2 (BMP-2), fibroblast growth factor 23 (FGF-23)-Klotho, alkaline phosphatase, vitamin K deficiency or vitamin K antagonists as well as endogenous inhibitors like Fetuin-A, matrix-Gla protein (MGP), osteopontin (OPN), osteoprotegerin (OPG), BMP-7, and pyrophosphate [27, 28, 19, 20, 29]. Osteo/chondrogenic transformed VSMC secrete inflammatory cytokines like tumor necrosis factor alfa (TNF-alfa), IL-1beta, IL-17A that in turn can also induce phenotypic VSMC transformation and consequent calcification via upregulation of Wnt/beta-catenin signaling pathway. Thus a vicious cycle is formed which contributes to the deepening of VC [30, 31]. Thus, altered VSMC phenotype and the imbalance between activators and inhibitors of vascular calcification create a microenvironment favoring the initiation and progression of VC.

Sincerely,

Bistra Galunska – corresponding author

Reviewer 3 Report

Comments and Suggestions for Authors

The manuscript entitled “Extrahepatic vitamin K-dependent Gla-proteins – potential cardiometabolic biomarkers” presents interesting issue but there are some problems which must be corrected before publishing.

Major:

The major problem is associated with the applied methodology. Authors did not specify if it was or not a systematic review – it should be clearly indicated. Based on the presented information it seems that it is not a systematic review, but some elements of a systematic review were applied.

If it was not a systematic review, it is the serious flaw of the presented manuscript, associated with the fact, that it presents a highly subjective review, not a systematic review. While the systematic review has a key role for broadening knowledge, the other reviews don’t have such role.

Abstract:

Authors should get familiar with the Instructions for authors (https://www.mdpi.com/journal/ijms/instructions) for the Abstract and correct this section. Within their Abstract Authors should present briefly not only background (justification for the study – as they presented), but also results and conclusions. Even if Authors present review article, it has its results (observations based on literature) and conclusions, which should be presented within this part of the manuscript.

Introduction:

Authors should present a broader perspective – not only for Europe and some information for United States, but international perspective based on the data obtained worldwide is needed.

Main body of the study:

Figures are the strong part of the study – I suppose that Authors of the manuscript prepared them themselves (as no reference is presented), but they could have been more attractive for reader (the current version – colours, design seems to be a little bit out of date, so may be not attractive for readers)

Limitations:

Authors should present strengths as well.

Materials and Methods:

Authors should broaden this section.

Conclusions:

Authors should deepen the practical implications of their study.

Author Response

BISTRA GALUNSKA

Department of Biochemistry, Molecular medicine and Nutrigenomics

Medical University of Varna

9000 Varna, Bulgaria

[email protected]

To

PROF. DR. DARIUSZ SZUKIEWICZ

Guest Editor

Special Issue “Molecular Mechanisms of Action of Adaptogens -

in Search of Natural Methods of Restoring Homeostasis”

MDPI

CC: Reviewer 3 of Manuscript ID: ijms-2901141

Type of manuscript: Review

Title: Extrahepatic vitamin K-dependent Gla-proteins – potential

cardiometabolic biomarkers

Authors: Bistra Galunska*, Yoto T Yotov, Miglena N Nikolova, Atanas A Atanasov

March 08, 2024

Dear Prof. Dr. Szukiewicz,

Dear Reviewer,

On 1-th March 2024, we were announced by email that our manuscript (ijms-2901141) entitled " Extrahepatic vitamin K-dependent Gla-proteins – potential cardiometabolic biomarkers", submitted for consideration for publication in the International Journal of Molecular Sciences, Special Issue “Molecular Mechanisms of Action of Adaptogens – in Search of Natural Methods of Restoring Homeostasis” as an review article has been reviewed.

We would like to thank the Reviewer for critical remarks and recommendations. All required revisions are included in the abstract, main text, and reference list. The changes we made in the manuscript are highlighted in yellow text.

Below are the detailed answers point by point to the reviewers’ comments. All answers and explanations are highlighted using bold and blue coloured text.

Comments and Suggestions for Authors

Reviewer 3

Major:

The major problem is associated with the applied methodology. Authors did not specify if it was or not a systematic review – it should be clearly indicated. Based on the presented information it seems that it is not a systematic review, but some elements of a systematic review were applied.

If it was not a systematic review, it is the serious flaw of the presented manuscript, associated with the fact, that it presents a highly subjective review, not a systematic review. While the systematic review has a key role for broadening knowledge, the other reviews don’t have such role.

Answer: Thank you for this comment. In the revised manuscript, we have clearly stated that this is a narrative review.

Abstract:

Authors should get familiar with the Instructions for authors (https://www.mdpi.com/journal/ijms/instructions) for the Abstract and correct this section. Within their Abstract Authors should present briefly not only background (justification for the study – as they presented), but also results and conclusions. Even if Authors present review article, it has its results (observations based on literature) and conclusions, which should be presented within this part of the manuscript.

Answer: The abstract has been thoroughly edited according to your recommendations including results and conclusions.

Abstract: One mechanism to regulate pathological vascular calcification (VC) is its active inhibition. Loss or inactivation of endogenic inhibitors is major inductor of VC. Such inhibitors are proteins rich in gamma-glutamyl residues (Gla-proteins), whose function strongly depends on vitamin K. The current narrative review is focused on discussing the role of extrahepatic vitamin K-dependent Gla-proteins (osteocalcin, OC; matrix Gla-protein, MGP; Gla-rich protein, GRP) in cardio-vascular pathology. Gla-proteins possess several functionally active forms whose role in the pathogenesis of VC is still unclear. It is assumed that low circulating non-phosphorylated MGP is indicator of active calcification and could be a novel biomarker of prevalent VC. High circulating completely inactive MGP is proposed as a novel risk factor for cardio-vascular events, disease progression, mortality, and vitamin K deficiency. The ratio between uncarboxylated (ucOC) and carboxylated (cOC) OC is considered as indicator of vitamin K status indirectly reflecting arterial calcium. Despite the evidence that OC is important energy metabolic regulator, its role on global cardio-vascular risk remains unclear. GRP acts as a molecular mediator between inflammation and calcification and may emerge a novel biomarker playing a key role in these processes. Gla-proteins benefit the clinical practice as inhibitors of VC, modifiable by dietary factors.

Introduction:

Authors should present a broader perspective – not only for Europe and some information for United States, but international perspective based on the data obtained worldwide is needed.

Answer: The introduction was thoroughly edited. At the beginning of the Introduction section, the following text marked in yellow including appropriate new references is inserted:

Cardiovascular diseases (CVD) are the leading cause of death and disability in Europe and worldwide. The leading diseases are coronary heart disease (CHD) and stroke. Deaths from CVD exceed those from all cancers in absolute numbers [1]. These diseases lead to excess premature mortality before the age of 70, with CVD accounting for over 60 million potentially lost young lives in Europe [2]. Ischemic heart disease (IHD) is the leading cause of death in 2023 in USA [3], as well as in other countries, such as Canada and New Zealand [4]. Every one in three deaths was due to these diseases in the United States in 2001-2011 [5], despite the 31% recorded reduction in mortality from atherosclerotic CVD. The situation in Europe is similar, with some regional variability. Since 1980, there has been a favorable downward trend in CVD mortality in Northern, Western and Southern Europe. In Central and Eastern Europe, however, the curve was horizontal or upward in different countries during the same time period [6]. The age-adjusted CVD mortality rate in the region is about twice the EU or economically advanced European average. For example, in Latvia and Romania, CVD mortality rates are 883 and 951/100,000 inhabitants, respectively, and are more than twice the EU average of 373.6/100,000 inhabitants. This is also true for premature mortality. In Russia and Belarus it is more than 10 times higher than in Switzerland - 300 vs. 26 per 100 000 inhabitants. CVD mortality rates for men aged 55-59 in some of the countries of the former Soviet Union are higher than those for men aged 75-79 in France [7]. The negative demographic trends with ageing of the population predetermine an increase in CVD morbidity and the associated health care costs [6]. For this reason, prevention of atherosclerotic CVD continues to be a major imperative for the societies. The Impact analysis of coronary heart disease (CHD) mortality trends in the United States from 1980 to 2000 indicates that at least 44% of the reduction in CVD mortality was due to risk factors (RFs) modification in the general population [8]. The same results were reported in other countries [9].

Main body of the study:

Figures are the strong part of the study – I suppose that Authors of the manuscript prepared them themselves (as no reference is presented), but they could have been more attractive for reader (the current version – colours, design seems to be a little bit out of date, so may be not attractive for readers)

Answer: We agree with the reviewer's comment that figures are presented in traditional style. Our goal regarding the figures was to complement the text in easiest and most understandable way.

Limitations:

Authors should present strengths as well.

Answer: This section was thoroughly edited and the strengths of the manuscript are pointed:

  1. Strengths and Limitations

We consider as a strength of the present review, that this is the first study analyzing the relation between statin use and circulating levels of different functional forms of extrahepatic vitamin K-dependent Gla proteins as endogenic inhibitors of vascular calcification.

However, this review has several limitations. First, due to lack of enough data, this review does not analyze the literature concerning possible effects of direct anticoagulants on the functional activity of Gla proteins, which is well known for vitamin K antagonists. Second, this review has not included epidemiologic data concerning possible additive effect of Gla proteins on the CVD risk profile. Moreover, data on OC and GRP expression at the cellular or tissue level are not include and analyzed.

Materials and Methods:

Authors should broaden this section.

Answer: As this is a narrative review, we have followed the criteria of SANRA specifying the search terms, inclusion criteria, and types of literature included [Baethge et al. Research Integrity and Peer Review (2019) 4:5. https://doi.org/10.1186/s41073-019-0064-8].

Conclusions:

Authors should deepen the practical implications of their study.

Answer: The conclusions were thoroughly edited and some practical aspects are pointed regarding:

  • the role of vitamin K supplements when indirect anticoagulants are used, which block the reactivation of vitamin K and gamma-glutamyl carboxylase and thus inhibit the conversion of inactive Gla-proteins into functionally active forms, which in turn provokes vascular calcification;
  • direct anticoagulants that do not affect the function of vitamin K-dependent gamma-carboxylases should be used if anticoagulant treatment is necessary;
  • the role of different functional forms of MGP as predictive and prognostic candidate biomarkers and as a therapeutic target for the treatment of vascular calcification and CVD.

Sincerely,

Bistra Galunska – corresponding author

Reviewer 4 Report

Comments and Suggestions for Authors

The manuscript is a review that addresses the topic of extrahepatic vitamin K-dependent Gla-proteins as potential cardiometabolic biomarkers. The authors focused on three vitamin K-depended proteins: osteocalcin (OC), matrix Gla-protein (MGP), and Gla-rich protein (GRPP.  The manuscript is well written in terms of the English language. The topic is interesting for researchers and clinicians, providing sufficient background to understand its message. However, it requires improvement by reviewing a few minor issues:

1.      The authors must mention the fact that MGP exercises its role locally (in tissues) and not systemically (in circulation), in comparison with the calciprotein particles that participate in the inhibitory activity of calcification in the blood.

2.      Lines 123-125: In order not to be misunderstood, the phrase must be reformulated, because it is suggested that both, gamma-carboxylation and phosphorylation, require vitamin K as a cofactor.

3.      In several places, the presentation of references is not appropriate: Line 213 (according to Mayers et al, not according to [49]); same at lines 393 and 560.

4.      References are missing in different places: line 574 (after “….from 21% to 43%”) and line 586 (after “A recent study on patients with proven CVD….”).

5.      There are abbreviations used without being first defined in the text (e.g. line 237: UA; line 463: HbA1c; line 467: CRP;  line 492: TC; line 493: TG )

6.      Try to be consistent with the abbreviations throughout the text (e.g. lines 239 and 258- UA instead of uric acid; line 400- OC instead of osteocalcin)

7.      There is a missing bracket at reference 54 in line 230.

8.      The word “some” does not sound scientific. Please replace it with “several” or another word of your choice (e.g. in line 263).

9.      Latin terms must be written in italics (e.g. in vitro, in vivo, ex vivo...). They appear in many places 9lines 372, 442, 449, 451, 489, 524, 531, 543, 606, 608, etc).

10.      In Chapter 6 (Limitations), please insert one more limitation of the review: the manuscript focused only on the changes in the circulating levels of OC, MGP, and GRP in humans, and not on their expression at the cellular or tissue level.

Overall, I enjoyed reading this well-organized and comprehensive review.

Comments on the Quality of English Language

Minor editing of English language required.

Author Response

BISTRA GALUNSKA

Department of Biochemistry, Molecular medicine and Nutrigenomics

Medical University of Varna

9000 Varna, Bulgaria

[email protected]

To

PROF. DR. DARIUSZ SZUKIEWICZ

Guest Editor

Special Issue “Molecular Mechanisms of Action of Adaptogens -

in Search of Natural Methods of Restoring Homeostasis”

MDPI

CC: Reviewer 4 of Manuscript ID: ijms-2901141

Type of manuscript: Review

Title: Extrahepatic vitamin K-dependent Gla-proteins – potential

cardiometabolic biomarkers

Authors: Bistra Galunska*, Yoto T Yotov, Miglena N Nikolova, Atanas A Atanasov

March 08, 2024

Dear Prof. Dr. Szukiewicz,

Dear Reviewer,

On 1-th March 2024, we were announced by email that our manuscript (ijms-2901141) entitled " Extrahepatic vitamin K-dependent Gla-proteins – potential cardiometabolic biomarkers", submitted for consideration for publication in the International Journal of Molecular Sciences, Special Issue “Molecular Mechanisms of Action of Adaptogens – in Search of Natural Methods of Restoring Homeostasis” as an review article has been reviewed.

We would like to thank the Reviewer for critical remarks and recommendations. All required revisions are included in the abstract, main text, and reference list. The changes we made in the manuscript are highlighted in yellow text.

Below are the detailed answers point by point to the reviewers’ comments. All answers and explanations are highlighted using bold and blue coloured text.

Comments and Suggestions for Authors

Reviewer 4

  1. The authors must mention the fact that MGP exercises its role locally (in tissues) and not systemically (in circulation), in comparison with the calciprotein particles that participate in the inhibitory activity of calcification in the blood.

Answer: In the section, “Matrix Gla-protein” is added the phrase that MGP is a “local tissue inhibitor” of vascular calcification.

  1. Lines 123-125: In order not to be misunderstood, the phrase must be reformulated, because it is suggested that both, gamma-carboxylation and phosphorylation, require vitamin K as a cofactor.

Answer: Thank you for this comment. Phrase reformulation has been made accordingly: MGP has been found to exist in 4 different conformations as a result of 2 post-translational modifications: serine phosphorylation and gamma-glutamate carboxylation, the latter of which requires vitamin K as a cofactor.

  1. In several places, the presentation of references is not appropriate: Line 213 (according to Mayers et al, not according to [49]); same at lines 393 and 560.
  2. References are missing in different places: line 574 (after “….from 21% to 43%”) and line 586 (after “A recent study on patients with proven CVD….”).
  3. There are abbreviations used without being first defined in the text (e.g. line 237: UA; line 463: HbA1c; line 467: CRP; line 492: TC; line 493: TG )
  4. Try to be consistent with the abbreviations throughout the text (e.g. lines 239 and 258- UA instead of uric acid; line 400- OC instead of osteocalcin)
  5. There is a missing bracket at reference 54 in line 230.
  6. The word “some” does not sound scientific. Please replace it with “several” or another word of your choice (e.g. in line 263).
  7. Latin terms must be written in italics (e.g. in vitro, in vivo, ex vivo...). They appear in many places 9lines 372, 442, 449, 451, 489, 524, 531, 543, 606, 608, etc).

Answers points 3-9: Thank you for the comments regarding technical aspects of manuscript preparation. All corrections concerning references, abbreviations, missing bracket, word “some”, and Latin terms have been made accordingly.

  1. In Chapter 6 (Limitations), please insert one more limitation of the review: the manuscript focused only on the changes in the circulating levels of OC, MGP, and GRP in humans, and not on their expression at the cellular or tissue level.

Answer: In section “5.6. Gene expression of MGP protein in peripheral blood mononuclear cells” of the manuscript the expression of MGP only in PBMCs is discussed, since these cells are available biological material obtained by minimally invasive procedures, which reflects tissue state. The fact that PBMCs are obtained by a minimally invasive procedure (venous blood sampling) is important as on the one hand, it is patient friendly and on the other, the result is informative. For this reason, only this aspect concerning gene expression of the discussed Gla-proteins is considered. For the other Gla proteins (OC and GRP)  subject of the present review, we found no evidence in the literature that they are expressed in PBMCs.

The corresponding text regarding the expression of other Gla-proteins at the cellular or tissue level has been added to the Limitation section: “Moreover, data on OC and GRP expression at the cellular or tissue level are not include and analyzed”.

Sincerely,

Bistra Galunska – corresponding author

Round 2

Reviewer 1 Report

Comments and Suggestions for Authors

UNFORTUNATELY. MOST OF MY COMMENTS WERE NOT ADDRESSED BY THE AUTHORS AND/OR NOT PROPERLY. SEE MY NEW COMMENTS INDICATED WITH -RR

  1. This is clearly a narrative review. Therefore, this point can be clearly stated and there is no need to discuss limitations, since the limitations of narrative reviews are well-know.

Answer: In the revised manuscript, we have clearly stated that this is a narrative review.

RR- ACCEPTED.

We consider that it is appropriate to point out some limitations of the review, as the literature data concerning some aspects of the role of Gla-proteins in vascular calcification are very limited and obviously cannot be discussed.

RR- AS MENTIONED, IT MAKES NO SENSE TO HAVE SUCH A SECTION SINCE THIS IS A NARRATIVE REVIEW AND THE LIMITATIONS ARE INTRINSIC. I STILL RECOMMEND THE AUTHORS TO REVISE THE MANUSCRIPT ACCORDINGLY.

  1. Similarly, it makes no sense to include a section for the methods, since this is not a systematic review.

Answer: Thank you for this comment. We believe that the inclusion of a section, indicating the main descriptors in literature search, (criteria for the selection of literature sources, keywords used, etc.) is useful.

RR- AS MENTIONED, IT MAKES NO SENSE TO HAVE SUCH A SECTION IN THEMETHODS, UNLESS THE AUTHORS DECIDE TO CHANGE THEIR REVIEW TO A SYSTEMATIC REVIEW, WHICH WOULD A COMPLETELY DIFFEERENT MANUSCRIPT AND THEN NEW SUBMISSION. I STILL RECOMMEND THE AUTHORS TO REVISE THE MANUSCRIPT ACCORDINGLY.

  1. A specific section for “perspectives/future direction” is appropriate, but should not be merged with the conclusion. In this section, the authors could also highlight and discuss the current gaps in the knowledge of this specific topic, which could be the “limitations” of the available evidence (rather than limitation of a narrative review)

Answer: The section “Future directions” is separated from section “Conclusions”. Current gaps in the literature regarding the role of discussed Gla-proteins in cardiometabolic health are pointed.

Future Directions. There is a bulk of evidence of the participation of vitamin K antagonists (VKA) in the vascular calcification. During the last decade, the use of direct oral anticoagulants (DOAC) is rising exponentially and is replacing the VKA for many indications. It is not clear how they will affect the Gla proteins and the process of vessel calcification.

The evidence for the univariate effect of the Gla-proteins in the CVD development are persuasive but there are not enough studies to prove that they retain their independent effect in multivariate analyses and to show their additive effect on the CVD risk profile.

RRR- THIS SECTION IS VERY APPROXIMATIVE AND SHOULD PROVIDE A DEEP ANALYSIS OF PERSPECTIVE. I RECOMMEND THE AUTHORS TO EXPAND THIS ANALYSIS.

  1. Conversely a conclusion section consisting of a paragraph is important to highlight the main relevance of these molecules according to the available literature.

Answer: The conclusion section has been edited accordingly. The inserted/edited text is highlighted in yellow.

  1. Conclusions

Vascular calcification is inextricably linked to atherosclerotic vascular disease, follows the course of the atherosclerotic process and also increases with age. It is no coincidence that the degree of calcium deposition in coronary vessels is perceived as a marker of the biological age of the vessels and the determination of CACS is a modern and up-to-date method for objective assessment of the risk of cardiovascular events.

In addition to the determination of CACS, which involves an invasive examination of the vascular status, additional non-invasive methods are being sought to assess the presence of atheromatous vascular deposits and especially their degree of calcification or to determine the risk of vascular calcification. A wide range of Gla-proteins, such as MGP, OC, and GRP are an attractive field of study as endogenic inhibitors of vascular calcification. A particular characteristic of these proteins is that their functionally active form is dependent on vitamin K. This is essential for the clinical practice, not only because of the widespread use of indirect anticoagulants as vitamin K antagonists, but also because the circulating levels of the active functional forms of Gla-proteins can be increased by dietary modification (food intake, vitamin K supplementation) or if anticoagulant treatment is necessary, it should be conducted with direct anticoagulants that do not affect the function of vitamin K-dependent gamma-carboxylases. Therefore, these Gla-proteins and especially different functional forms of MGP are being thoroughly investigated as predictive and prognostic candidate biomarkers and as a therapeutic target for the treatment of vascular calcification and CVD.

RRR- THE CONCLUSION IS DEFINITELY TOO LONG AND, THEREFORE, DOES NOT HIGHLIGHT THE MOST IMPORTANT POINTS. MOREOVER, SOME CONCEPTS MAY BE A STARTING POINT FOR THE PREVIOUS SECTION, IN MY OPINION.

  1. I would also suggest to add a section discussion discussing the role of Gla-proteins in other diseases. In this additional section, a mention to COVID-19 should be also done, considering the predisposing role of CVD for severe forms of this disease. Therefore, it would be interesting to check and discuss if there is any evidence in this regard, as well. This article (e.g. The Pleiotropic Role of Vitamin K in Multimorbidity of Chronic Obstructive Pulmonary Disease. J Clin Med. 2023 Feb 5;12(4):1261. doi: 10.3390/jcm12041261) and others may provide useful insights in this regard. Moreover, a recent paper reported that MGP can facilitate CD8+ T cell exhaustion by activating the NF-κB pathway (see: Int J Biol Sci. 2022 Mar 6;18(6):2345-2361. doi: 10.7150/ijbs.70137) and CD8+ T cell subpopulation balance/homeostasis has been shown to be significantly perturbated in COVID-19 patients (refer to: Respir Res. 2022 Oct 10;23(1):278. doi: 10.1186/s12931-022-02190-8).

Answer: Thank you for this recommendation and for extremely interesting papers. At this stage, the focus of our research and of this review is the interrelationship between the functional activity of extrahepatic vitamin K-dependent Gla proteins and vascular calcification as a pathogenic element of cardiovascular events and associated risk factors. For this reason, the role of these proteins in other diseases, including SARS-CoV infection, has not been discussed. As the extrahepatic vitamin K-dependent Gla proteins are subject of intensive studies in relation to their pleiotropic effects and the possibility of modifying their activity by vitamin K supplementation, we are ready to prepare another review article on the role of extrahepatic Gla proteins in other diseases beyond CVD, which would be of considerable interest to the scientific community.

RR- THE AUTHORS HAVE NOT ADDRESSED AT ALL THIS RECOMMENDATION. I SUGGEST THE AUTHORS TO RECONSIDER THIS POINT, WHICH MAY BE ALSO ANOTHER POINT TO BE ADDRESSED AT LEAST IN THE PERSPECTIVE OR FUTURE DIRECTION SECTION.

  1. “There are currently 17 known types of vitamin K-dependent proteins, including blood coagulation factors, matrix Gla-protein (MGP), growth arrest specific protein 6 (Gas6), anticoagulant proteins C, S and Z, osteocalcin (OC), Gla residue-rich protein (GRP), periostin (isoforms 1 and 4), periostin-like factor (PLF), proline-rich Gla proteins (PRGP 1 and 2), transmembrane Gla proteins (TMG3 and TMG4). Only 14 of them have been identified in humans [8, 9, 10].” The authors mentioned that there are at least 14 vit. K dependent-proteins expressed in humans. However, specific sections were dedicated to few of them. The authors should clarify why only these were described in detail and, in general, at the end of the introduction should better explain the objective of this narrative review.

Answer: We appreciate your recommendation. The following text is inserted at the end of the Introduction: “Among the multiple inhibitors of vascular calcification the subject of this review are the extrahepatic Gla proteins, whose functional activity strongly depends on the vitamin K level. This is important because vitamin K status appears to be a modifying factor with respect to the inhibitory activity of these Gla proteins. Improving vitamin K status by diet or vitamin K supplements, would contribute to reducing the risk of vascular calcification and associated with it adverse outcomes for cardiovascular health. Furthermore, it should be taken into account that when anticoagulant therapy is needed, in order to preserve the inhibitory activity of these vitamin K-dependent Gla-proteins, direct anticoagulants that do not affect the function of vitamin K-dependent gamma-carboxylases are recommended instead of vitamin K antagonists”.

RRR- I THINK THE AUTHORS SHOULD CLEARLY STATE THAT ONLY SOME OF VIT-K DEPENDENT PROTEINES WERE ANALYZED IN THIS REVIEW AND CLEARLY EXPLAIN THE REASON, WHICH IS NOT VERY WELL EXPLAINED IN MY OPINION.

Comments on the Quality of English Language

SEE ABOVE

Author Response

BISTRA GALUNSKA

Department of Biochemistry, Molecular medicine and Nutrigenomics

Medical University of Varna

9000 Varna, Bulgaria

[email protected]

To

PROF. DR. DARIUSZ SZUKIEWICZ

Guest Editor

Special Issue “Molecular Mechanisms of Action of Adaptogens -

in Search of Natural Methods of Restoring Homeostasis”

MDPI

CC: Reviewer 1 of Manuscript ID: ijms-2901141

Type of manuscript: Review

Title: Extrahepatic vitamin K-dependent Gla-proteins – potential

cardiometabolic biomarkers

Authors: Bistra Galunska*, Yoto T Yotov, Miglena N Nikolova, Atanas A Atanasov

March 15, 2024

Dear Prof. Dr. Szukiewicz,

Dear Reviewer,

On 13-th March 2024, we were announced by email that we havr to provide ansvers and to correct the manuscript based on reviewer 1's comments (Round 2).

We would like to thank the Reviewer for the critical remarks and recommendations. All required revisions are included in the main text, and reference list. The changes we made in the manuscript are highlighted in yellow text.

Below are the detailed answers point by point to the reviewers’ comments (Round 2). All answers and explanations are highlighted using bold.

Comments and Suggestions for Authors

Reviewer 1

  1. This is clearly a narrative review. Therefore, this point can be clearly stated and there is no need to discuss limitations, since the limitations of narrative reviews are well-know.

Answer: In the revised manuscript, we have clearly stated that this is a narrative review.

RR- ACCEPTED.

We consider that it is appropriate to point out some limitations of the review, as the literature data concerning some aspects of the role of Gla-proteins in vascular calcification are very limited and obviously cannot be discussed.

RR- AS MENTIONED, IT MAKES NO SENSE TO HAVE SUCH A SECTION SINCE THIS IS A NARRATIVE REVIEW AND THE LIMITATIONS ARE INTRINSIC. I STILL RECOMMEND THE AUTHORS TO REVISE THE MANUSCRIPT ACCORDINGLY.

Answer 2-nd Round:

The section “Limitations” is now removed from the manuscript.

  1. Similarly, it makes no sense to include a section for the methods, since this is not a systematic review.

Answer: Thank you for this comment. We believe that the inclusion of a section, indicating the main descriptors in literature search, (criteria for the selection of literature sources, keywords used, etc.) is useful.

RR- AS MENTIONED, IT MAKES NO SENSE TO HAVE SUCH A SECTION IN THE METHODS, UNLESS THE AUTHORS DECIDE TO CHANGE THEIR REVIEW TO A SYSTEMATIC REVIEW, WHICH WOULD A COMPLETELY DIFFEERENT MANUSCRIPT AND THEN NEW SUBMISSION. I STILL RECOMMEND THE AUTHORS TO REVISE THE MANUSCRIPT ACCORDINGLY.

Answer 2-nd Round:

As this is a narrative review, we have followed the criteria of SANRA specifying the search terms, inclusion criteria, and types of literature included [Baethge et al. Research Integrity and Peer Review (2019) 4:5. https://doi.org/10.1186/s41073-019-0064-8].

“A convincing narrative review will be transparent about the sources of information on which the text is based”…...”narrative review to refer to an attempt to summarize the literature in a way which is not explicitly systematic, where the minimum requirement for the term systematic relates to the method of the literature search, but in a wider sense includes a specific research question and a comprehensive summary of all studies”.

In order to implement the reviewer's recommendation we have removed the section “Methods” from the manuscript.

  1. A specific section for “perspectives/future direction” is appropriate, but should not be merged with the conclusion. In this section, the authors could also highlight and discuss the current gaps in the knowledge of this specific topic, which could be the “limitations” of the available evidence (rather than limitation of a narrative review)

Answer: The section “Future directions” is separated from section “Conclusions”. Current gaps in the literature regarding the role of discussed Gla-proteins in cardiometabolic health are pointed.

Future Directions. There is a bulk of evidence of the participation of vitamin K antagonists (VKA) in the vascular calcification. During the last decade, the use of direct oral anticoagulants (DOAC) is rising exponentially and is replacing the VKA for many indications. It is not clear how they will affect the Gla proteins and the process of vessel calcification.

The evidence for the univariate effect of the Gla-proteins in the CVD development are persuasive but there are not enough studies to prove that they retain their independent effect in multivariate analyses and to show their additive effect on the CVD risk profile.

RRR- THIS SECTION IS VERY APPROXIMATIVE AND SHOULD PROVIDE A DEEP ANALYSIS OF PERSPECTIVE. I RECOMMEND THE AUTHORS TO EXPAND THIS ANALYSIS.

Answer 2-nd Round:

This section has been extensively edited. We have tried to do a more thorough analysis of future perspectives by including the possible role of the Gla proteins, the subject of this review, in other diseases, including COVID 19. The following text along with new references is inserted:

“8. Future directions

The degree of calcium deposition in coronary vessels is perceived as a marker of the biological age of the vessels and the determination of CACS is a modern and up-to-date method for objective assessment of the risk of cardiovascular events. In addition to the determination of CACS, which involves an invasive examination of the vascular status, a wide range of Gla-proteins, such as MGP, OC, and GRP are an attractive field of study as endogenic inhibitors of vascular calcification. Demonstration of a causal relationship between the levels of different functional forms of these Gla-protein and CACS, as well as its validation in large cohorts of patients with CVD would be the rationale for their future inclusion as non-invasive molecular biomarkers to assess the presence of atheromatous vascular deposits and especially their degree of calcification or to determine the risk of vascular calcification. In this aspect, different functional forms of MGP are being thoroughly investigated as predictive and prognostic candidate biomarkers and as a therapeutic target for the treatment of vascular calcification and CVD.

A particular characteristic of these proteins is that their functionally active forms are dependent on vitamin K. Proving the diagnostic utility of different functional forms of these Gla-proteins will enable future studies that could change the diagnostic algorithm by introducing circulating vitamin K-dependent Gla-proteins to other biomarkers already established and used in practice, as well as to change the treatment algorithm by including vitamin K supplementation of CVD patients.

Moreover, modulation of extrahepatic Gla-proteins’ activity by vitamin K is essential for the clinical practice when anticoagulant treatment is necessary as should be conducted with direct anticoagulants (DOAC) that do not affect the function of vitamin K-dependent gamma-carboxylases and the activation of these endogenic VC inhibitors. As it is not yet studied how DOAC will affect different functional forms of Gla proteins and the process of vessel calcification, this is another important aspect of future studies. As the uc-dpMGP and ucOC/cOC ratio has been found to reflect the body's extrahepatic functional vitamin K status, validation of these parameters in large cohort studies would allow to evaluate the effects of vitamin K supplementation on cardiovascular outcomes. This will subsequently contribute to the development of an adequate program to monitor functional vitamin K status and assess the need for supplementation with vitamin K2 or long-chain menaquinones to reduce the complications of prevalent CVD.

Another point of future studies that is still obscure and needs further clarification, is related to possible molecular mechanisms through which statins, a widely used medications in the treatment atherosclerosis and coronary heart disease, may affect vitamin K status, the activity of vitamin K dependent Gla-proteins and their functions involved in vascular protection.

Regarding the link with CVD risk profile, the evidence for the univariate effect of the Gla-proteins in the CVD development are persuasive, but future studies are needed to prove that they retain their independent effect in multivariate analyses and to show their additive effect on the CVD risk profile.

Аs vascular calcification is involved in the pathogenesis of numerous diseases, it could be assumed that active forms of vitamin K-dependent Gla proteins also play a role in other diseases beyond CVD. Matrix Gla proteins may be useful biomarkers in other pathological conditions. Although there are some data on their presence in patients with Type 2 diabetes mellitus [172] Adel H, Fawzy O, Mahmoud E, Mohammed NS, Khidr EG. Inactive matrix Gla protein in relation to diabetic retinopathy in type 2 diabetes. J Diabetes Metab Disord. 2023; 22(1): 603-610. doi: 10.1007/s40200-022-01180-3.; [173] Xie FF, Zhang YF, Hu YF, Xie YY, Wang XY, Wang SZ, Xie BQ. Significance of serum glucagon-like peptide-1 and matrix Gla protein levels in patients with diabetes and osteoporosis. World J Clin Cases. 2022; 10(5): 1527-1535. doi: 10.12998/wjcc.v10.i5.1527., the research is sparse in people with Type 1 diabetes. Those patients are at excessive risk of micro- and macrovascular complications and of early atherosclerosis and calcification of the arteries.

The COVID-19 pandemic gave a boost of the research in the pathogenesis of vascular reactions and alterations in infectious diseases. The vit. K metabolism and depletion during COVID-19, together with the decrease in the vit K-dependent activation of MGP leaves the elastic fibres in the hepar unprotected against the infection, with increased thrombogenicity [174] Janssen R, Visser MPJ, Dofferhoff ASM, Vermeer C, Janssens W, Walk J. Vitamin K metabolism as the potential missing link between lung damage and thromboembolism in Coronavirus disease 2019. British Journal of Nutrition. 2021;126(2):191-198. doi:10.1017/S0007114520003979. Future efforts should be concentrated on the pathophysiology of the MGP in other acute infections, as well as in more chronic conditions, like tuberculosis. It may reveal their role in the development and adaptation of the immune system.

Initial studies demonstrate high dp-ucMGP levels in COPD patients. Vit K deficiency leads to loss of elastin in lung tissues [175] Piscaer I, van den Ouweland JMW, Vermeersch K, Reynaert NL, Franssen FME, Keene S, Wouters EFM, Janssens W, Vermeer C, Janssen R. Low Vitamin K Status Is Associated with Increased Elastin Degradation in Chronic Obstructive Pulmonary Disease. J Clin Med. 2019; 8(8): 1116. doi: 10.3390/jcm8081116. There is a correlation between lung emphysema and vit. K status, with increased levels of dp-ucMGP and diffuse capacity for carbon monoxide. Epidemiologic studies reveal higher mortality in patients with COPD in the upper quartile of dp-ucMGP. [176] Piscaer I, Janssen R, Franssen FME, Schurgers LJ, Wouters EFM. The Pleiotropic Role of Vitamin K in Multimorbidity of Chronic Obstructive Pulmonary Disease. J Clin Med. 2023; 12(4): 1261. doi: 10.3390/jcm12041261. However, the exact mechanisms and the diagnostic work-up are not still very clear and need future research.

Another possible application of the MGPs are in chronic inflammatory diseases. There are preliminary studies in rheumatoid arthritis [177] Ghorbanihaghjo A, Hajialilo M, Shahidi M, Khabazi A, Kolahi S, Reza Jafari Nakhjavani M, Raeisi S, Argani H, Rashtchizadeh N. Osteoprotegerin (OPG) and Matrix Gla protein (MGP) in rheumatoid arthritis patients: Relation to disease activity. Egypt Rheumatol 2014; 36(3): 111-116. https://doi.org/10.1016/j.ejr.2014.01.003., chronic inflammatory colitis, and others. Future perspective is more extensive research in autoimmune diseases and other chronic inflammation conditions which may reveal the potential role of vit K related proteins in the pathologic processes.”

  1. Conversely a conclusion section consisting of a paragraph is important to highlight the main relevance of these molecules according to the available literature.

Answer: The conclusion section has been edited accordingly. The inserted/edited text is highlighted in yellow.

  1. Conclusions

Vascular calcification is inextricably linked to atherosclerotic vascular disease, follows the course of the atherosclerotic process and also increases with age. It is no coincidence that the degree of calcium deposition in coronary vessels is perceived as a marker of the biological age of the vessels and the determination of CACS is a modern and up-to-date method for objective assessment of the risk of cardiovascular events.

In addition to evaluating CACS, which involves an invasive examination of the vascular status, additional non-invasive methods including biomarkers are being sought to assess the presence of atheromatous vascular deposits and especially their degree of calcification or to determine the risk of vascular calcification. A wide range of Gla-proteins, such as MGP, OC, and GRP are an attractive field of study as endogenic inhibitors of vascular calcification. A particular characteristic of these proteins is that their functionally active form is dependent on vitamin K. This is essential for the clinical practice, not only because of the widespread use of indirect anticoagulants as vitamin K antagonists, but also because the circulating levels of the active functional forms of Gla-proteins can be increased by dietary modification (food intake, vitamin K supplementation) or if anticoagulant treatment is necessary, it should be conducted with direct anticoagulants that do not affect the function of vitamin K-dependent gamma-carboxylases. Therefore, these The Gla-proteins and especially different functional forms of MGP are being thoroughly investigated as predictive and prognostic candidate biomarkers and as a therapeutic target for the treatment of vascular calcification and CVD and with a potential for other pathologic conditions.

RRR- THE CONCLUSION IS DEFINITELY TOO LONG AND, THEREFORE, DOES NOT HIGHLIGHT THE MOST IMPORTANT POINTS. MOREOVER, SOME CONCEPTS MAY BE A STARTING POINT FOR THE PREVIOUS SECTION, IN MY OPINION.

Answer 2-nd Round:

This section has been edited as recommended.

  1. Conclusions

Vascular calcification is inextricably linked to atherosclerotic vascular disease, follows the course of the atherosclerotic process and also increases with age. In addition to evaluating CACS, non-invasive methods including biomarkers are being sought to assess the presence of atheromatous vascular deposits and especially their degree of calcification or to determine the risk of vascular calcification. A wide range of Gla-proteins, such as MGP, OC, and GRP are an attractive field of study as endogenic inhibitors of vascular calcification. A particular characteristic of these proteins is that their functionally active form is dependent on vitamin K. This is essential for the clinical practice, not only because of the widespread use of indirect anticoagulants as vitamin K antagonists, but also because the circulating levels of the active functional forms of Gla-proteins can be increased by dietary modification (food intake, vitamin K supplementation) or if anticoagulant treatment is necessary, it should be conducted with direct anticoagulants that do not affect the function of vitamin K-dependent gamma-carboxylases. The Gla-proteins and especially different functional forms of MGP are being thoroughly investigated as predictive and prognostic candidate biomarkers and as a therapeutic target for the treatment of vascular calcification and CVD and with a potential for other pathologic conditions.

  1. I would also suggest to add a section discussion discussing the role of Gla-proteins in other diseases. In this additional section, a mention to COVID-19 should be also done, considering the predisposing role of CVD for severe forms of this disease. Therefore, it would be interesting to check and discuss if there is any evidence in this regard, as well. This article (e.g. The Pleiotropic Role of Vitamin K in Multimorbidity of Chronic Obstructive Pulmonary Disease. J Clin Med. 2023 Feb 5;12(4):1261. doi: 10.3390/jcm12041261) and others may provide useful insights in this regard. Moreover, a recent paper reported that MGP can facilitate CD8+ T cell exhaustion by activating the NF-κB pathway (see: Int J Biol Sci. 2022 Mar 6;18(6):2345-2361. doi: 10.7150/ijbs.70137) and CD8+ T cell subpopulation balance/homeostasis has been shown to be significantly perturbated in COVID-19 patients (refer to: Respir Res. 2022 Oct 10;23(1):278. doi: 10.1186/s12931-022-02190-8).

Answer: Thank you for this recommendation and for extremely interesting papers. At this stage, the focus of our research and of this review is the interrelationship between the functional activity of extrahepatic vitamin K-dependent Gla proteins and vascular calcification as a pathogenic element of cardiovascular events and associated risk factors. For this reason, the role of these proteins in other diseases, including SARS-CoV infection, has not been discussed. As the extrahepatic vitamin K-dependent Gla proteins are subject of intensive studies in relation to their pleiotropic effects and the possibility of modifying their activity by vitamin K supplementation, we are ready to prepare another review article on the role of extrahepatic Gla proteins in other diseases beyond CVD, which would be of considerable interest to the scientific community.

RR- THE AUTHORS HAVE NOT ADDRESSED AT ALL THIS RECOMMENDATION. I SUGGEST THE AUTHORS TO RECONSIDER THIS POINT, WHICH MAY BE ALSO ANOTHER POINT TO BE ADDRESSED AT LEAST IN THE PERSPECTIVE OR FUTURE DIRECTION SECTION.

Answer 2-nd Round:

Thank you for this comment. It is addressed in the Future directions section. – in diabetes, infectious diseases, COPD, chronic inflammatory diseases, and COVID 19. We do not think that separate section discussion is needed in a narrative review.

  1. “There are currently 17 known types of vitamin K-dependent proteins, including blood coagulation factors, matrix Gla-protein (MGP), growth arrest specific protein 6 (Gas6), anticoagulant proteins C, S and Z, osteocalcin (OC), Gla residue-rich protein (GRP), periostin (isoforms 1 and 4), periostin-like factor (PLF), proline-rich Gla proteins (PRGP 1 and 2), transmembrane Gla proteins (TMG3 and TMG4). Only 14 of them have been identified in humans [8, 9, 10].” The authors mentioned that there are at least 14 vit. K dependent-proteins expressed in humans. However, specific sections were dedicated to few of them. The authors should clarify why only these were described in detail and, in general, at the end of the introduction should better explain the objective of this narrative review.

Answer: We appreciate your recommendation. The following text is inserted at the end of the Introduction: “Among the multiple inhibitors of vascular calcification the subject of this review are the extrahepatic Gla proteins, whose functional activity strongly depends on the vitamin K level. This is important because vitamin K status appears to be a modifying factor with respect to the inhibitory activity of these Gla proteins. Improving vitamin K status by diet or vitamin K supplements, would contribute to reducing the risk of vascular calcification and associated with it adverse outcomes for cardiovascular health. Furthermore, it should be taken into account that when anticoagulant therapy is needed, in order to preserve the inhibitory activity of these vitamin K-dependent Gla-proteins, direct anticoagulants that do not affect the function of vitamin K-dependent gamma-carboxylases are recommended instead of vitamin K antagonists”.

RRR- I THINK THE AUTHORS SHOULD CLEARLY STATE THAT ONLY SOME OF VIT-K DEPENDENT PROTEINES WERE ANALYZED IN THIS REVIEW AND CLEARLY EXPLAIN THE REASON, WHICH IS NOT VERY WELL EXPLAINED IN MY OPINION.

Answer 2-nd Round:

Thank you for this comment. We edited the corresponding text and hope that now more clearly is stated why only MGP, OC, and GRP of vit.-K dependent Gla-proteins were analyzed in the current review.

Among the multiple inhibitors of vascular calcification the subject of this review are the extrahepatic Gla proteins, whose functional activity strongly depends on the vitamin K levels. This is important because vitamin K status appears to be a modifying factor with respect to the inhibitory activity of these Gla proteins. Thereof а subject of the present review are only those extrahepatic vitamin K-dependent Gla proteins that are related to ectopic calcification such as vascular calcification. More in-depth studies and data in the literature are found for osteocalcin, matrix Gla-protein, Gla-rich protein.

Sincerely,

Bistra Galunska – corresponding author

Round 3

Reviewer 1 Report

Comments and Suggestions for Authors

Overall, the previous comments were sufficiently addressed. 

Comments on the Quality of English Language

see above